# PASA: A Principled Embedding-Space Watermarking Approach for LLM-Generated Text under Semantic-Invariant Attacks

**Zhenxin Ai** [1]   **Haiyun He** [1] [*]

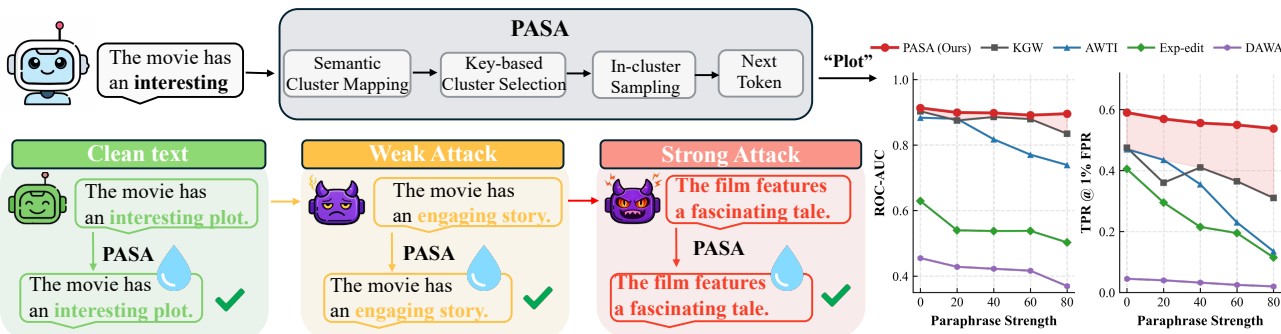

*Figure 1.* **Left:** Illustration of **PASA**, a principled watermarking approach operating in the **latent embedding space** on semantic clusters. By anchoring **shared randomness** to semantic clusters via a secret key, PASA remains robust against **semantic-invariant attacks** (e.g., paraphrasing) while ensuring **distortion-free** generation. **Right:** Quantitative results demonstrating that PASA outperforms standard vocabulary-space watermarking baselines across varying paraphrase strengths in both AUC-ROC and TPR@1%FPR.

## Abstract

Watermarking for large language models (LLMs) is a promising approach for detecting LLM-generated text and enabling responsible deployment. However, existing watermarking methods are often vulnerable to semantic-invariant attacks, such as paraphrasing. We propose PASA, a principled, robust, and distortion-free watermarking algorithm that embeds and detects a watermark at the semantic level. PASA operates on semantic clusters in a latent embedding space and constructs a distributional dependency between token and auxiliary sequences via shared randomness synchronized by a secret key and semantic history. This design is grounded in our theoretical framework that characterizes a jointly optimal embedding-detection pair, achieving the fundamental trade-offs among detection accuracy, robustness, and distortion. Evaluations across multi-

ple LLMs and semantic-invariant attacks demonstrate that PASA remains robust even under strong paraphrasing attacks while preserving high text quality, outperforming standard vocabulary-space baselines. Ablation studies further validate the effectiveness of our hyperparameter choices. Web-page: PASA.

## 1. Introduction

Transformer-based large language models (LLMs) have demonstrated remarkable fluency and coherence in open-ended generation (Achiam et al., 2024; Touvron et al., 2023; Yang et al., 2025a). As LLMs become increasingly powerful, the distinction between machine-generated and human-authored text has become blurred. This raises significant concerns around misuse, including large-scale disinformation (Vykopal et al., 2024; Zhu et al., 2025b), automated spear phishing and targeted deception (Hazell, 2023), amplified threats to organizational security (Mirsky et al., 2023), and challenges to academic evaluation systems (Balalle & Pannilage, 2025).

These concerns motivate the need for verifiable provenance and accountable attribution. Recent work has focused on active provenance via LLM watermarking (Kirchenbauer et al., 2023; Liu et al., 2024c; Yang et al., 2025b; Dathathri

[1] The Hong Kong University of Science and Technology (Guangzhou), Guangzhou, China. Zhenxin Ai: zhenxinai@hkust-gz.edu.cn . Correspondence to: Haiyun He <haiyunhe@hkust-gz.edu.cn>.

*Proceedings of the 43$^{rd}$ International Conference on Machine Learning*, Seoul, South Korea. PMLR 306, 2026. Copyright 2026 by the author(s).

et al., 2024), which operates directly in the generation process. Unlike post-hoc detectors that are often unreliable, black-box watermarking leverages secret-key–conditioned randomized sampling to insert imperceptible yet statistically detectable patterns into generated text. This mechanism enables reliable third-party detection using only the text, without requiring access to model parameters or APIs.

However, most existing watermarking schemes operate directly on the token vocabulary and construct detection statistics over surface-level token identities. Consequently, such approaches are inherently vulnerable to **semantic-invariant attacks**: meaning-preserving transformations, such as synonym substitution or paraphrasing, can arbitrarily alter the token realization while leaving the underlying semantics intact. As a result, semantic-invariant rewriting may easily remove the token-level watermarks and distort the associated detection statistics, undermining the effectiveness of naive watermarking schemes. While some alternatives improve robustness via heuristic semantic-aware logit biases (Fu et al., 2024b; Guo et al., 2024; He et al., 2024), they inevitably shift the token distribution in expectation and sacrifice text quality for detectability.

This observation highlights a fundamental scientific challenge: *can we design a watermarking method that balances the following three facets?* (i) **Robustness** under semantic-preserving transformations, (ii) **Distortion-free** generation, in the sense of preserving the original generation distribution, and (iii) **Principled control over detection errors**, particularly at low false-positive (false-alarm) rates, under adversarial semantic perturbations.

Inspired by the well-known green/red list watermarking paradigm (Kirchenbauer et al., 2023), early attempts seek to improve robustness by aligning watermark behavior with contextual embeddings, i.e., token representations that depend on surrounding context, via soft mappings (Liu et al., 2024a; Zhang et al., 2024b). Along this line, subsequent studies further refine token-level logit biases (watermarking rules) to better trade off robustness and text quality (Giboulot & Furon, 2024; Shen et al., 2025; Kirchenbauer et al., 2024). For instance, Liu & Bu (2024) employs an adaptive embedding strategy guided by token entropy, together with semantic-based seeding to mitigate quality degradation while enhancing robustness. These methods aim to better reflect semantic similarity than raw token identities, but still operate largely at the token level. More recently, partition-and-constrain strategies have been explored to design watermarking schemes related to semantic representations. SemStamp (Hou et al., 2024a) and k-SemStamp (Hou et al., 2024b) partition the sentence-embedding space using locality-sensitive hashing (LSH) or clustering to define watermark regions, while CoheMark (Zhang et al., 2025a) leverages fuzzy clustering to encourage discourse-level con-

sistency. These results suggest that the geometric structure of the **latent semantic space** can provide a more stable anchor for watermarking than raw tokens. However, these approaches are largely heuristic and do not offer principled guarantees on the trade-offs among robustness, distortion, and detection accuracy. In parallel, some theoretical efforts have explored the fundamental trade-off between distortion and detection accuracy from both optimization and statistical viewpoints (Takezawa et al., 2023; Wouters, 2024; Cai et al., 2024; Huang et al., 2023; Li et al., 2025). For example, DAWA (He et al., 2025) proves the optimality of a distribution-adaptive approach at the token level, paired with a model-agnostic detector, achieving high true-positive rates (TPRs) at ultra-low false-positive rates (FPRs). Nonetheless, these works fail to incorporate robustness into their frameworks, nor do they guide the principled design of robust watermarking schemes. For a more comprehensive literature review, please refer to Appendix B.

Taken together, existing approaches reveal a clear gap between practice and theory in LLM watermarking. On the one hand, semantic-aware designs suggest that operating in latent embedding spaces can substantially improve robustness to semantic-invariant attacks. On the other hand, existing theoretical frameworks primarily focus on token-level watermarking and do not account for robustness under meaning-preserving transformations, leaving the fundamental trade-offs among robustness, distortion, and detection accuracy poorly understood. This gap motivates a principled watermarking framework that operates at the semantic level while offering explicit theoretical guarantees.

In this work, we introduce **PASA**, a **P**rincipled watermarking **A**pproach under **S**emantic-invariant **A**ttacks, which bridges this gap by elevating watermarking from the token level to the semantic level within a formal theoretical framework (cf. Figure 1). **PASA** operates in a latent semantic embedding space, embedding and detecting watermarks through a carefully designed distributional dependency between token sequences and auxiliary random sequences. Semantic-level shared randomness is synchronized by a secret key and the semantic history of a context window. Concretely, PASA models semantic-invariant rewriting through a semantic mapping function that assigns tokens to semantic clusters in the latent space, and introduces a novel two-stage sampling mechanism that enables stringent control of false alarms while maintaining distortion-free generation. This design is grounded in an information-theoretic framework extended from (He et al., 2025) that characterizes the jointly optimal embedding–detection pair at the sequence level, achieving strong detection accuracy and semantic robustness while strictly preserving the original distribution.

Our contributions can be summarized as follows:

- We propose PASA, a principled watermarking method

that operates within the latent semantic space rather than on individual tokens. By anchoring shared randomness at the semantic level, PASA achieves superior detection performance and distortion-free generation while remaining robust to semantic-invariant text modifications.

- We provide a theoretical framework for robust watermark embedding and detection under semantic-invariant attacks, which grounds the design of PASA. Within this framework, we characterize the fundamental trade-offs among detection accuracy, robustness, and distortion, and identify the jointly optimal embedding-detection pair for a given attack model, providing formal guarantees for PASA.
- Extensive evaluations across multiple models and datasets demonstrate that PASA consistently outperforms existing baselines under T5-based replacement and DIPPER paraphrasing attacks. Results confirm superior detectability at low FPRs without compromising text quality or computational efficiency.

## 2. A Theoretical Framework for Robust and Distortion-Free Watermarking

In this section, we develop a theoretical framework for designing robust and distortion-free watermark embedding and detection schemes for LLM-generated text, and formalize a semantic-invariant attack model.

**Next-Token-Prediction (NTP) Distribution.** LLMs generate text token by token in an auto-regressive way. A token is the basic processing unit of an LLM and typically corresponds to a word fragment in natural languages. Let $\mathcal{V}$ denote the token vocabulary, with size $|\mathcal{V}| = \mathcal{O}(10^4)$ (Liu, 2019; Radford et al., 2019; Zhang et al., 2022; Touvron et al., 2023). At each step $t$, given a prompt pt and the previous tokens $x^{t-1}$, an *unwatermarked* LLM samples the next token $X_t$ according to a Next-Token-Prediction (NTP) distribution $Q_t := Q_{X_t|x^{t-1},\text{pt}}$. This induces a joint distribution of a length-$T$ token sequence $X^T = (X_1, \ldots, X_T)$, given by $Q_{X^T} = \prod_{t=1}^T Q_{X_t|X^{t-1}}$. We assume that a well-behaved unwatermarked LLM is distributionally indistinguishable from human text generation, and therefore also treat $Q_t$ as the human NTP distribution. For notational simplicity, the dependence on the prompt pt is suppressed.

**Watermark Embedding.** In this paper, we adopt the theoretical framework for LLM watermark embedding from He et al. (2025), which encompasses most existing in-process sampling-based watermarking schemes. The watermark embedding scheme constructs an *auxiliary random sequence* $\zeta^T \sim P_{\zeta^T}$ drawn from a space $\mathcal{Z}^T$, and a *dependence structure* between $\zeta^T$ and the token sequence $X^T$. Therefore, given the auxiliary sequence $\zeta^T$, the *watermarked* LLM samples the next token $X_t$ according to a modified NTP distribution $P_{X_t|x^{t-1},\zeta_t}$, and the induced

conditional joint distribution of token sequence is given by $P_{X^T|\zeta^T} = \prod_{t=1}^T P_{X_t|X^{t-1},\zeta^T}$. Note that the joint distribution of the watermarked token sequence $X^T$ is given by $P_{X^T}$, which might be different from the original $Q_{X^T}$.

We define a watermark embedding scheme as $\epsilon$-distorted if the statistical divergence between the watermarked distribution $P_{X^T}$ and the original $Q_{X^T}$ satisfies

$$\mathsf{D}(P_{X^T}, Q_{X^T}) \le \epsilon,$$

where D can be any distortion metric that measures the dissimilarity between distributions. For $\epsilon = 0$, the watermark embedding scheme is *distortion-free*.

**Watermark Detection under Semantic-Invariant Attacks.** A common randomness is shared through the auxiliary random sequence $\zeta^T$ and a secret key between the embedding and detection phases. If a watermarked LLM generates a token sequence $X^T$, it depends on $\zeta^T$ statistically; otherwise, $X^T$ and $\zeta^T$ are independent. The watermark detection thus boils down to a binary hypothesis testing problem:

- $\mathrm{H}_0$: $X^T$ is generated by a human, i.e., $(X^T, \zeta^T) \sim Q_{X^T} \otimes P_{\zeta^T}$;
- $\mathrm{H}_1$: $X^T$ is generated by a watermarked LLM, i.e., $(X^T, \zeta^T) \sim P_{X^T,\zeta^T}$.

However, the detector may receive watermarked text that has been altered by an adversary. We consider a broad class of semantic-invariant attacks, where the text can be modified in arbitrary ways as long as its semantics are preserved, such as token replacement and paraphrasing. Specifically, let $f : \mathcal{V}^T \to [K]$ be a surjective function that maps a token sequence $X^T$ to $K$ distinct semantic clusters in the latent embedding space. Clearly, given any token sequence $x^T$, $f$ induces an equivalence class containing $x^T$: $\mathcal{B}_f(x^T) := \{\tilde{x}^T \in \mathcal{V}^T : f(\tilde{x}^T) = f(x^T)\}$. Assuming that the adversary can arbitrarily modify any token sequence $x^T$ within its equivalence class $\mathcal{B}_f(x^T)$, we evaluate a detector $\gamma : \mathcal{V}^T \times \mathcal{Z}^T \to \{0, 1\}$ by its worst-case detection errors over all possible attacks induced by $f$:

- False-alarm (FA) error:

$$\beta_0^f(\gamma, Q_{X^T}, P_{\zeta^T}) := \mathbb{E}_{Q_{X^T} \otimes P_{\zeta^T}} \left[ \sup_{\tilde{x}^T \in \mathcal{B}_f(X^T)} \gamma(\tilde{x}^T, \zeta^T) \right].$$

- Miss-detection (MD) error:

$$\beta_1^f(\gamma, P_{X^T,\zeta^T}) := \mathbb{E}_{P_{X^T,\zeta^T}} \left[ \sup_{\tilde{x}^T \in \mathcal{B}_f(X^T)} (1 - \gamma(\tilde{x}^T, \zeta^T)) \right].$$

FA error occurs when human-written text is detected as watermarked, whereas MD error occurs when watermarked LLM-generated text is classified as human-written.

**Optimization Problem.** As human behaviors may vary widely, to effectively reduce the FA error in reality, we aim to control the *worst-case* FA error over all possible

human texts under a threshold $\alpha \in (0,1)$. Our objective is to design a robust and $\epsilon$-distorted watermark embedding scheme and detector that minimizes the MD error while controlling the worst-case FA error, namely, solving the optimization problem

$$\inf_{\gamma, P_{X^T, \zeta^T}} \beta_1^f(\gamma, P_{X^T, \zeta^T}) \tag{P}$$

$$\text{s.t. } \sup_{Q_{X^T}} \beta_0^f(\gamma, Q_{X^T}, P_{\zeta^T}) \leq \alpha, \ \mathsf{D}(P_{X^T}, Q_{X^T}) \leq \epsilon.$$

Here, we allow the distortion level $\epsilon \geq 0$ to demonstrate the trade-off among the MD error $\beta_1^f$, the FA constraint $\alpha$, the size of the output set of $f$, and $\epsilon$. However, in practice, we enforce $\epsilon = 0$ for a distortion-free watermarking approach.

## 3. Theoretical Foundations and Algorithm

Building on the semantic-invariant attack model $f$ formalized in the framework, we develop the theoretical foundations of robust watermarking and derive an algorithm that leverages semantic representations to embed watermarks in the latent embedding space.

### 3.1. Theoretical Foundations

**Error-Robustness-Distortion Trade-Offs.** We characterize the fundamental trade-offs among the detection errors, robustness level, and distortion level by presenting the optimal objective value of the optimization problem (P) in the following theorem. In particular, the robustness level of a watermarking scheme is inversely related to the size $K$ of the semantic cluster set induced by the semantic mapping function $f$.

**Theorem 1** (Minimum MD Error). *Given any tuple of* $(Q_{X^T}, \alpha, \epsilon, f)$*, the minimum MD error attained from* (P) *is*

$$\beta_1^{f,*} := \min_{\substack{P_{X^T}: \\ \mathsf{D}(P_{X^T}, Q_{X^T}) \leq \epsilon}} \sum_{k \in [K]} \left( \left( \sum_{x^T : f(x^T) = k} P_{X^T}(x^T) \right) - \alpha \right)_+.$$

The proof of Theorem 1 is deferred to Appendix C. The characterization immediately reveals that the minimum MD error $\beta_1^{f,*}$ decreases as the distortion level $\epsilon$ or the FA constraint $\alpha$ increases, and as the robustness requirement is relaxed (i.e., as $K$ increases). In the extreme case $K = |\mathcal{V}|$, the result reduces to the classical setting in which robustness is not incorporated into the watermarking design.

**Jointly Optimal Robust and Distortion-Free Scheme.** We derive the jointly optimal watermark embedding and detection schemes that achieve the minimum MD error $\beta_1^{f,*}$. In particular, we let $\epsilon = 0$ and thus $P_{X^T} = Q_{X^T}$, leading to a distortion-free scheme.

**Theorem 2** ((Informal) Jointly Optimal Watermark Embedding and Detection). *The optimal pair of watermark*

detector and embedding method accepts the form:

- *Detector:*
$$\gamma^*(X^T, \zeta^T) = \mathbb{1}\{f(X^T) = \mathsf{vec2num}(\zeta^T)\}, \tag{1}$$
*where* $\mathsf{vec2num} : \mathcal{Z}^T \to [K] \cup \{\tilde{\zeta}\}$ *is a bijective function that maps a sequence to a real number and* $\tilde{\zeta} \in \mathbb{N} \setminus [K]$ *is called the overflow state.*

- *Embedding method: the watermark embedding consists of two stages: 1) construct the auxiliary sequence distribution* $P_{\zeta^T}^*$*; 2) construct the conditional sampling distribution* $P_{X^T | \zeta^T}^*$ *associated with* $\gamma^*$*, such that* $\mathbb{E}_{\zeta^T}[P_{X^T | \zeta^T}^*] = Q_{X^T}$*. The detailed expressions are presented in the algorithm design below.*

The formal statement and proof of Theorem 2 is deferred to Appendix D. The optimal design embeds and detects watermarks in the latent semantic embedding space induced by the attack model $f$, aligning with intuitive semantic invariance. Specifically, the optimal auxiliary distribution $P_{\zeta^T}^*$ is a "truncated" version of the semantic embedding distribution, augmented with an overflow state $\tilde{\zeta}$ to control the FA error. Conditioned on the sampled auxiliary sequence $\zeta^T$, the resulting conditional sampling distribution performs a re-normalized in-cluster token sampling, and preserves the original token sequence distribution $Q_{X^T}$ in expectation. These theoretical insights directly motivate our practical algorithm design.

### 3.2. Algorithm Design

In this section, we introduce a **P**rincipled embedding-space watermarking **A**pproach under **S**emantic-invariant **A**ttacks (**PASA**). Building on the theoretical foundations and insights, PASA embeds a watermark into LLM-generated text in the latent token embedding space via a two-stage sampling strategy according to Theorem 2, while preserving the original NTP distribution. For detection, PASA accumulates the score $\mathbb{1}\{f(x_t) = \zeta_t\}$ for a given $\zeta_t \in [K] \cup \{\tilde{\zeta}\}$ (cf. (1)) across tokens $x_t$ and compares it to a threshold. This approach achieves high detection accuracy under semantic-invariant attacks while preserving text generation quality.

3.2.1. WATERMARK EMBEDDING VIA A TWO-STAGE SAMPLING STRATEGY

We implement the embedding method proven in Theorem 2 at each token generation step $t$.

**Stage 1: Auxiliary Distribution Construction and Sampling.** As shown in Figure 2, we first construct a surjective mapping $f : \mathcal{V} \to [K]$, partitioning the token embedding space into $K$ disjoint semantic clusters.

**(G1) Semantic Cluster Distribution.** The semantic mapping function $f$ directly transforms the NTP distribution $Q_t$

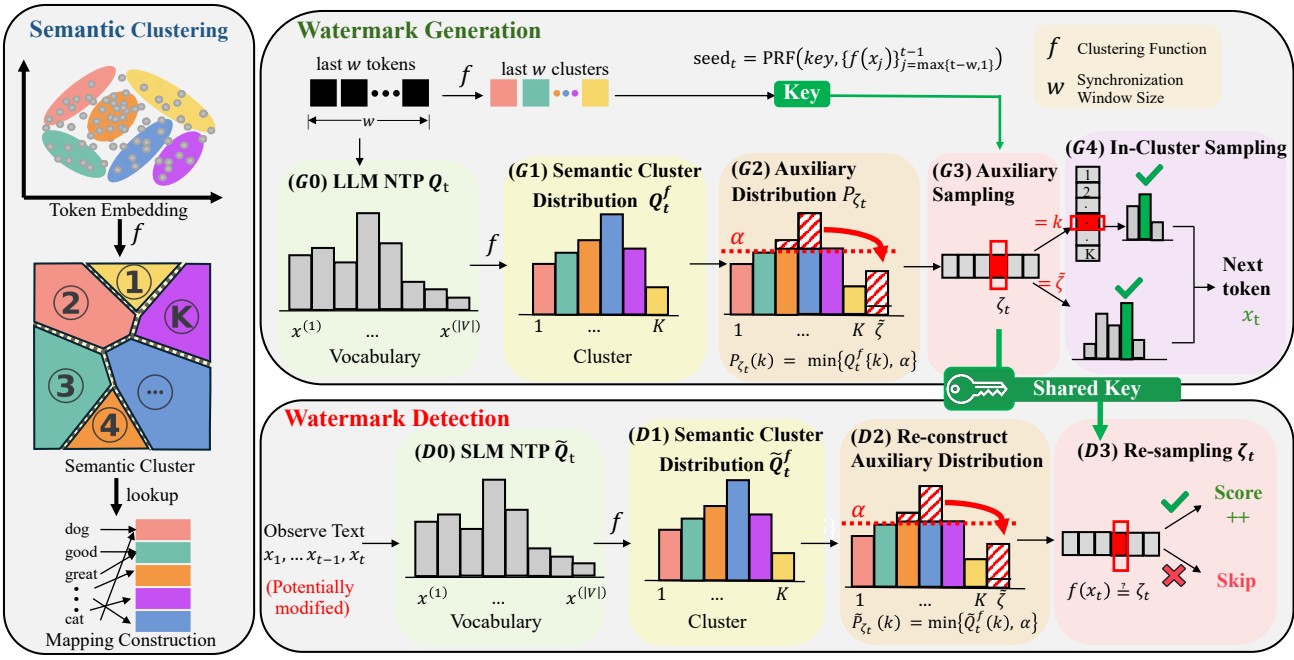

*Figure 2.* **Overview of PASA. Left:** Construction of the semantic mapping function $f$, which partitions the latent token embedding space into $K$ semantic clusters. **Right: Top (Generation).** **(G1)** At each step $t$, the NTP distribution $Q_t$ is transformed into the cluster distribution $Q_t^f$. **(G2)** The auxiliary distribution $P_{\zeta_t}$ is truncated by a threshold $\alpha$ and contains an overflow state $\tilde{\zeta}$ to ensure FA error control. **(G3)** Auxiliary sampling of $\zeta_t$ uses a $\mathsf{seed}_t$ generated by a PRF with a secret key and $w$ semantic history as input. **(G4)** The sampled auxiliary random variable $\zeta_t$ guides the sampling of the next token $x_t$ within the selected semantic cluster. **Bottom (Detection).** **(D0-D2)** For a potentially modified observed token sequence, the detector approximates the generation distribution through an SLM. **(D3)** The detection score accumulates based on the alignment between the resampled $\zeta_t$ and the observed semantic cluster $f(x_t)$.

to a semantic cluster distribution:
$$Q_t^f(k) := \sum_{x:f(x)=k} Q_t(x), \quad \forall k \in [K],$$
which is insensitive to the token-level perturbation.

**(G2) Auxiliary Distribution.** We construct the auxiliary distribution $P_{\zeta_t}$ on the latent space $\mathcal{Z} = [K] \cup \{\tilde{\zeta}\}$ w.r.t. the semantic cluster distribution $Q_t^f$, where $\tilde{\zeta}$ represents the overflow state (cf. Theorem 2). Given a FA error constraint $\alpha$, we let $P_{\zeta_t}(k) = \min\{Q_t^f(k), \alpha\}$ for all semantic cluster index $k \in [K]$, and accumulate the overflowed probability masses to the overflow state $\tilde{\zeta}$:
$$P_{\zeta_t}(\tilde{\zeta}) = 1 - \sum_{k=1}^{K} P_{\zeta_t}(k) = \sum_{k=1}^{K} (Q_t^f(k) - \alpha)_+.$$
This construction ensures that the MD error is minimized while the FA error is controlled under $\alpha$, as shown in the proof of Theorem 2.

**(G3) Auxiliary Sampling.** We sample the auxiliary variable $\zeta_t \sim P_{\zeta_t}$ using a seed generated by a pseudo-random function (PRF), whose input consists of the semantic cluster indices of the previous $w$ tokens and a shared secret key:
$$\mathsf{seed}_t = \mathrm{PRF}(\mathsf{key}, \{f(x_j)\}_{j=\max\{t-w,1\}}^{t-1}).$$
The seeds can be recovered during detection with the shared secret key and the semantic mapping function.

**Stage 2: (G4) In-Cluster Sampling.** The next token is sampled according to the constructed sampling distribution $P_{X_t|x^{t-1},\zeta_t}$ conditioned on the auxiliary variable $\zeta_t$. Given different values of sampled $\zeta_t$, the next token sampling proceeds via two branches

- If $\zeta_t = k \in [K]$, we sample $X_t$ within the semantic cluster $k$ according to a re-normalized distribution:
$$X_t \sim \left( \frac{Q_t(x)\mathbb{1}\{f(x) = k\}}{Q_t^f(k)} \right)_{x \in \mathcal{V}}.$$

- If $\zeta_t = \tilde{\zeta}$, we sample $X_t$ within each semantic cluster $k$ with a probability proportional to the overflow mass $(Q_t^f(k) - \alpha)_+$, which maintains the NTP distribution identical to $Q_t$ in expectation. The conditional sampling distribution over tokens $x \in \mathcal{V}$ is given by
$$P_{X_t|x^{t-1},\zeta_t}(x) \propto (Q_t^f(f(x)) - \alpha)_+ \frac{Q_t(x)}{Q_t^f(f(x))}.$$

This two-stage sampling strategy enables semantic-level watermark embedding and ensures distortion-free generation where $\mathbb{E}_{\zeta_t}[P_{X_t|x^{t-1},\zeta_t}] = Q_t$, while allowing the detector to recover the auxiliary sequence via a shared secret key.

*Table 1.* **Detection performance on clean text and under semantic-invariant token-replacement attacks.** Comparisons of ROC-AUC, TPR@1%FPR, and TPR@10%FPR across Llama2-13B and Mistral-8×7B architectures. T5-Large and T5-XXL are used as attackers. **Best**, Second Best, and Third Best results are marked in each column.

| LLM | Method | Clean Text | | | Token Replacement Attack | | | | | |
| | | C4 | | | T5-Large | | | T5-XXL | | |
| | | ROC-AUC↑ | TPR@1%FPR↑ | TPR@10%FPR↑ | ROC-AUC↑ | TPR@1%FPR↑ | TPR@10%FPR↑ | ROC-AUC↑ | TPR@1%FPR↑ | TPR@10%FPR↑ |
|---|---|---|---|---|---|---|---|---|---|---|
| Llama2-13B | KGW+23 | 0.9990 | 0.9950 | 0.9950 | 0.9750 | 0.7350 | 0.9300 | 0.9196 | 0.6800 | 0.8100 |
| | Exp-edit | **1.0000** | **1.0000** | **1.0000** | 0.9905 | 0.9400 | 0.9700 | 0.9536 | **0.8700** | **0.9250** |
| | AWTI | **1.0000** | **1.0000** | **1.0000** | **0.9978** | **0.9978** | **0.9950** | **0.9702** | 0.7950 | 0.9050 |
| | DAWA | 0.9950 | 0.9950 | 0.9950 | 0.8067 | 0.3300 | 0.5700 | 0.6280 | 0.1450 | 0.2900 |
| | Ours | **1.0000** | **1.0000** | **1.0000** | 0.9946 | 0.9296 | 0.9899 | 0.9392 | 0.8040 | 0.8794 |
| Mistral-8×7B | Exp-edit | 0.9993 | 0.9950 | 0.9950 | 0.9760 | 0.9250 | 0.9650 | 0.9409 | **0.8750** | **0.9100** |
| | AWTI | **1.0000** | **1.0000** | **1.0000** | **0.9972** | **0.9400** | **0.9900** | **0.9611** | 0.7200 | 0.8900 |
| | DAWA | **1.0000** | **1.0000** | **1.0000** | 0.8696 | 0.5350 | 0.6800 | 0.7397 | 0.2750 | 0.4300 |
| | Ours | **1.0000** | **1.0000** | **1.0000** | 0.9902 | 0.9300 | 0.9700 | 0.9222 | 0.7650 | 0.8600 |

*Table 2.* **Detection performance under semantic-invariant paraphrasing attacks (DIPPER).** Results are reported for three configurations with increasing structural perturbation (Order Diversity), ranging from $Ord = 0$ to $Ord = 80$, with fixed lexical diversity $Lex = 60$. **Best**, Second Best, and Third Best results are marked in each column.

| Method | DIPPER (Lex=60, Ord=0) | | | DIPPER (Lex=60, Ord=20) | | | DIPPER (Lex=60, Ord=80) | | |
| | ROC-AUC↑ | TPR@1%FPR↑ | TPR@10%FPR↑ | ROC-AUC↑ | TPR@1%FPR↑ | TPR@10%FPR↑ | ROC-AUC↑ | TPR@1%FPR↑ | TPR@10%FPR↑ |
|---|---|---|---|---|---|---|---|---|---|
| KGW+23 | 0.9032 | 0.4750 | 0.7000 | 0.8750 | 0.3600 | 0.6400 | 0.8346 | 0.3100 | 0.5200 |
| Exp-edit | 0.6291 | 0.4050 | 0.5000 | 0.5398 | 0.2950 | 0.4050 | 0.5029 | 0.1150 | 0.3150 |
| AWTI | 0.8835 | 0.4700 | 0.6850 | 0.8801 | 0.4350 | 0.6800 | 0.7391 | 0.1350 | 0.3600 |
| DAWA | 0.4551 | 0.0450 | 0.1300 | 0.4284 | 0.0400 | 0.0800 | 0.3697 | 0.0200 | 0.0400 |
| **Ours** | **0.9132** | **0.5900** | **0.7970** | **0.8993** | **0.5693** | **0.7744** | **0.8956** | **0.5377** | **0.7593** |

### 3.2.2. WATERMARK DETECTION

The detector observes a token sequence $x^T$ and has access to the shared semantic mapping function $f$, the secret key, the FA error constraint $\alpha$, and a surrogate language model (SLM). The SLM, with NTP distribution denoted by $\tilde{Q}_t$, is a lightweight and parameter-efficient approximation of the LLM suitable for local deployment and facilitates detection. The detection process mirrors the generation procedure at each token position $t$.

**(D0) & (D1) Approximation.** With the SLM, the detector obtains an approximated NTP distribution $\tilde{Q}_t$ for each token $x_t$ and transforms it to the corresponding semantic cluster distribution $\tilde{Q}_t^f$ via the semantic mapping function $f$.

**(D2) Reconstruct Auxiliary Distribution.** Similar to (G2) in the watermark embedding process, the detector reconstructs the auxiliary distribution $\tilde{P}_{\zeta_t}$ based on the approximate $\tilde{Q}_t^f$ and the threshold $\alpha$.

**(D3) Replay and Scoring.** With the shared secret key and the observed semantic history $\{f(x_j)\}_{j=\max\{t-w,1\}}^{t-1}$, the detector recovers the seed $\text{seed}_t$ with the same PRF and re-samples $\zeta_t \sim \tilde{P}_{\zeta_t}$. Grounded by Theorem 2, the detector accumulates the score $\mathbb{1}\{f(x_t) = \zeta_t\}$ for each observed pair $(x_t, \zeta_t)$. When the re-sampled $\zeta_t$ matches the semantic cluster of $x_t$, the token contributes a unit score; when

they do not match or $\zeta_t = \tilde{\zeta}$, the token is skipped since $\mathbb{1}\{f(x_t) = \zeta_t\} \equiv 0$. Notably, this mechanism allows the detector to skip some low-entropy tokens with certain probabilities, which effectively reduces the FA error in practice.

## 4. Experiments

This section presents an empirical evaluation of our proposed PASA algorithm.

### 4.1. Experimental Setup

**Semantic Mapping and Clustering.** We adopt a pretrained model `gte-Qwen2-7B-instruct` (Li et al., 2023) to encode each token as a semantic embedding vector in the latent space. To ensure semantic consistency, we embed tokens using a fixed instruction template and apply $\ell_2$ normalization, so that similarity in the latent space is measured by cosine similarity. We then apply K-means clustering (Lloyd, 1982) to partition the embedding space into $K$ disjoint semantic clusters (setting $K = 4$ by default), thereby defining the semantic mapping function $f$.

**Models and Dataset.** We implement PASA on Llama-2-13B (Touvron et al., 2023) and Mixtral-8×7B (Jiang et al., 2023). For black-box detection, we use smaller proxy SLMs (Llama-2-7B and Mistral-7B, respectively). All experiments

are conducted on `realnewslike` from C4 (Raffel et al., 2020). We additionally evaluate generalization on the long-form QA dataset `ELI5` (see Appendix A).

**Attacks.** We evaluate robustness under two semantic-invariant paradigms: (i) contextual token replacement using T5-Large/T5-XXL (Raffel et al., 2020) with mask ratio $r \in \{0.3, 0.5\}$; (ii) paraphrasing using DIPPER (Krishna et al., 2023) with three intensities by varying lexical and word-order diversity $(L, O)$. Detailed configurations and hyperparameters are provided in Appendix F.

**Evaluation Metrics.** We report AUROC and TPR at low FPR (e.g., TPR@1%FPR). We compare against KGW (Kirchenbauer et al., 2023), Exp-Edit (Kuditipudi et al., 2024), AWTI (Liu & Bu, 2024), and DAWA (He et al., 2025). We evaluate text quality via PPL using a fixed `GPT-NeoX-20B` evaluator (Black et al., 2022), and report average generation/detection latency per sample.

### 4.2. Main Results

**Clean and Token-Replacement Detection.** Table 1 summarizes the detection performance on clean and modified text under T5-based token-level replacement. On clean text, PASA achieves near-perfect detection accuracy across both Llama-2-13B and Mixtral-8×7B, validating its effectiveness in non-adversarial settings.

Under T5-based attacks, standard schemes such as KGW and DAWA degrade substantially due to the sensitivity of token identities. In contrast, PASA maintains competitive stability. Specifically, under the T5-Large attack on Llama-2-13B, PASA achieves a TPR@1%FPR of 0.9296, significantly outperforming KGW (0.7350) and DAWA (0.3300). Even under the more aggressive T5-XXL attack, PASA maintains an AUROC of 0.9392, exceeding KGW and matching robust baselines like Exp-edit. On the sparse Mixtral-8×7B model, PASA surpasses Exp-edit under the T5-Large attack with an AUROC of 0.9902. These results confirm that anchoring randomness within the latent semantic space mitigates the state mismatch induced by local perturbations, thereby enhancing watermark survivability.

**Robustness against Paraphrasing.** Table 2 presents detection performance under DIPPER paraphrasing attacks, where lexical substitution is fixed at $\text{LEX} = 60$ and word-order perturbation increases across $\text{ORD} \in \{0, 20, 80\}$. Larger ORD indicates stronger syntactic reordering, yielding a more challenging semantic-invariant attack. Compared to token-replacement attacks, DIPPER paraphrases induce broader structural variation, which amplifies the performance gap between semantic-level methods and approaches relying on token-level statistics. PASA achieves the most robust detection at low FPRs across all settings.

For $\text{ORD} = 0, 20$, and $80$, it attains TPR@1%FPR at 0.5578, 0.5829, and 0.5879, with corresponding AUROCs of 0.8776, 0.9116, and 0.8934. In contrast, token-level baselines degrade sharply under paraphrasing. In particular, at $\text{ORD} = 80$, the TPR1@%FPR of DAWA drops to only 0.0200 and KGW to only 0.3050. Moreover, methods designed specifically for robust editing, Exp-edit and AWTI, also deteriorate under strong reordering, with TPR@1%FPR dropping to 0.1150 and 0.1350, respectively. Overall, these results indicate that synchronizing shared randomness at the semantic level enables PASA to better withstand the desynchronization induced by meaning-preserving paraphrases, especially word reordering, thereby improving watermark robustness. Further robustness studies and comparisons with representative robust watermarking methods are presented in Appendix A, which consistently support the above observations.

*Table 3.* **Comparison of generation quality and computational efficiency.** We report Perplexity (PPL) on GPT-NeoX-20B (Black et al., 2022) to validate the **distortion-free** property, alongside the average Generation Time and Detection Time per sample.

| Method | PPL | Gen. Time (s) | Det. Time (s) |
|---|---|---|---|
| Human | 10.41 | – | – |
| Unwatermark | 12.41 | 12.93 | – |
| KGW+23 | 11.81 | – | **0.04** |
| Exp-edit | 23.40 | – | 2.41 |
| AWTI | 19.77 | 24.24 | 10.52 |
| DAWA | **8.41** | 13.56 | 0.31 |
| **Ours** | 11.44 | **13.35** | 0.27 |

### 4.3. Quality and Efficiency

**Text Quality.** Table 3 provides a comparative analysis of the perplexity (PPL) for text generated by PASA relative to the unwatermarked baseline and prior methods, including KGW, DAWA, Exp-edit, and AWTI. Theoretically, the generation mechanism of PASA strictly preserves the per-step NTP distribution of the underlying model in expectation, ensuring the output is distortion-free. To provide a more concrete illustration of the generated content, we also present several examples of PASA-watermarked text in Appendix E.

The empirical results are consistent with this theoretical guarantee. PASA achieves a PPL of 11.44, which remains close to the unwatermarked baseline (12.41) and human text (10.41), indicating that the watermark introduces little degradation to generation quality. PASA also achieves comparable perplexity to KGW (11.81), while substantially outperforming Exp-edit (23.40) and AWTI (19.77), both of which exhibit much higher PPL values. Although DAWA reports the lowest PPL (8.41), PASA maintains a text quality statistically closer to the original model than DAWA. Overall, these results confirm that PASA effectively preserves

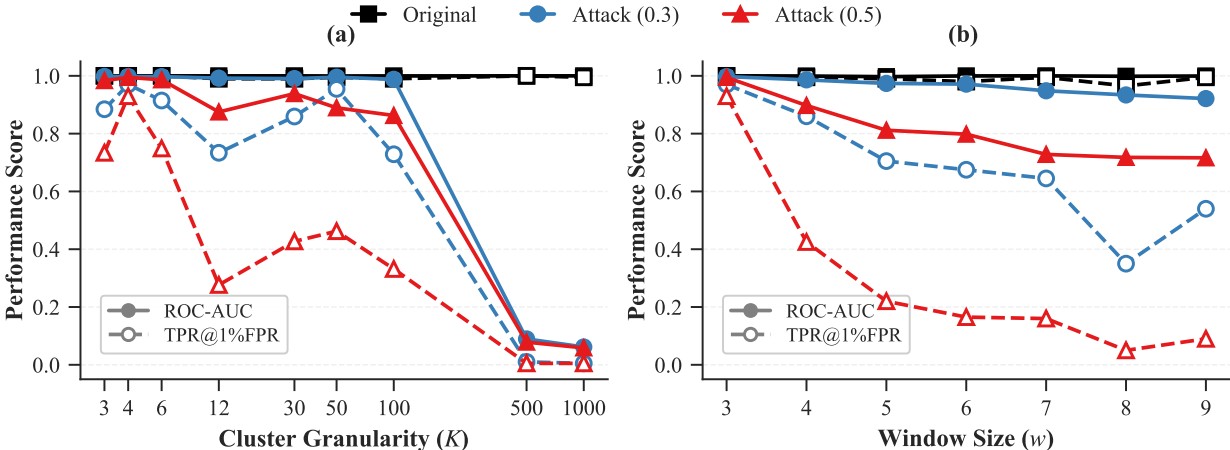

Figure 3. **Ablation study on hyper-parameters.** (a) Impact of semantic cluster granularity ($K$) on robustness across log-scale cluster counts. (b) Impact of synchronization window size ($w$) on robustness. The plots compare the baseline (Original) against T5-based token replacement attacks ($r = 0.3, 0.5$).

generation quality and adheres to the distortion-free property. Importantly, PASA remains effective for long-form generations as well; see Appendix A.

**Computational Efficiency.** To quantify runtime overhead, we measure average latency for watermarked generation relative to an unwatermarked baseline. For each configuration, we generate 200 sequences of fixed length 300 tokens. As reported in Table 3, PASA incurs a marginal increase in generation latency ($12.93\,s \rightarrow 13.35\,s$; $< 0.5\,s$). This suggests that the cost of semantic clustering and distribution computation is negligible compared to autoregressive decoding. PASA is also efficient relative to prior methods. Its generation latency is comparable to DAWA ($13.56\,s$) and substantially lower than AWTI ($24.24\,s$). For detection, PASA achieves the lowest latency ($0.27\,s$), outperforming Exp-edit ($2.41\,s$) and AWTI ($10.52\,s$). We note that KGW achieves the fastest detection (0.04s) due to its simple token-level, count-based detector with negligible computation. While lightweight designs are brittle under semantic-invariant edits, PASA maintains low detection latency without sacrificing robustness. Although runtimes are not directly comparable in Table 3 due to pipeline differences, separate matched evaluations show that Exp-edit requires approximately $1.5\times$ the total runtime of PASA. Overall, PASA remains computationally efficient and practical for deployment.

### 4.4. Ablation Study

**Semantic Cluster Granularity.** Our proposed PASA algorithms rely on the defined semantic mapping function $f$ that partitions the embedding space into $K$ clusters. Both theoretically (cf. Theorem 1) and empirically, $K$ determines the robustness level and governs the trade-off be-

tween robustness and detection accuracy. Figure 3 illustrates detection accuracy across a range of $K$, showing that PASA remains robust with coarse-to-moderate partitions while robustness degrades when the partition becomes excessively fine-grained. First of all, we notice that detection on clean text remains near-perfect for all $K$, which means that the choice of $K$ only affects the robustness of PASA. For $K \in [3, 100]$, robustness against T5-based token-replacement attacks is well preserved. Empirically, we observe that $K = 4$ achieves the best overall performance across all evaluation metrics, and we therefore adopt $K = 4$ in our experimental setting. However, as $K$ exceeds 500, PASA's detection accuracy on modified text degrades rapidly. This degradation arises from two aspects: 1) as predicted by the fundamental robustness-detection accuracy trade-off in Theorem 1, finer semantic clustering causes semantic-level watermarking to gradually revert back to token-level behavior; 2) the random seed generation becomes more fragile: since the seed is derived from the semantic clusters of previous tokens, finer clustering increases sensitivity to perturbations, making seed recovery less reliable under attack.

**Synchronization Window Size.** We further examine the influence of the synchronization window size $w$, which determines how much recent semantic context is used to generate the seed to sample an auxiliary random sequence. As shown in Figure 3, robustness exhibits an inverse relationship with $w$, revealing a trade-off between contextual aggregation and synchronization stability. Under severe token-replacement attacks at rate $r = 0.5$, as the window expands from $w = 3$ to $w = 8$, TPR@1%FPR decreases from 0.7236 to 0.1508. This indicates that a longer semantic context window increases seed sensitivity to token perturbations and thus impacts the detection accuracy under attacks.

However, an overly small $w$ may impair generated text coherence, since the generation pseudo-randomness is only determined by a small set of semantic cluster combinations.

Overall, these ablation studies help identify suitable hyperparameters that strike a sweet spot between robustness and text quality, yielding semantic clusters that are coarse enough to remain invariant under paraphrasing while being sufficiently local to maintain seed synchronization.

## 5. Conclusion

In this paper, we presented PASA, a principled watermarking algorithm that operates in a latent embedding space to enhance the robustness of LLM watermarking against semantic-invariant attacks. By partitioning the latent space into disjoint semantic clusters and employing a sampling mechanism synchronized by a secret key and an auxiliary random sequence, PASA establishes shared randomness at the semantic level, which is the key to its robustness during detection. Our design is grounded in a theoretical characterization that identifies a jointly optimal embedding-detection pair at the sequence level, revealing the fundamental trade-off among detection accuracy and robustness. We also prove that this approach is distortion-free, as it strictly preserves the model's original generation distribution. Extensive experiments, including cross-model evaluations, demonstrate that PASA maintains robust detectability against token replacement and paraphrasing attacks without compromising text quality. These findings validate that a principled semantics-aware design greatly improves the effectiveness of LLM watermarking, suggesting directions for further improving robustness and enhancing generalization across diverse generative models.

## 6. Limitations

**Limitations.** Our method still has several limitations. First, PASA may degrade under very strong rewriting or watermark-removal attacks that substantially change both the semantic content and the distributional structure of the text. Incorporating richer contextual or sentence-level semantics may further improve robustness, but would also increase modeling complexity.

Second, detection on very short texts remains challenging. Since our detector aggregates token-level statistical evidence, short sequences provide fewer observations and therefore weaker detection confidence. This issue may be alleviated by combining token-level evidence with sentence-level or passage-level statistics.

Third, PASA is most effective when the detector-side SLM is tokenizer-compatible with the generation model. Tokenizer mismatch can weaken the consistency between watermark embedding and detection, reducing cross-family transferability. A practical solution is to deploy multiple lightweight SLM detectors from different candidate model families, where a high-confidence response from one detector can verify the watermark and suggest the likely source model family.

## Acknowledgments

This work was supported by the Guangdong Provincial Key Lab of Integrated Communication, Sensing and Computation for Ubiquitous Internet of Things (No. 2023B1212010007).

## Impact Statement

As generative models become deeply integrated into society, the ability to distinguish machine-generated text from human-authored content is essential for mitigating misinformation, ensuring academic integrity, and protecting intellectual property. Implementing watermarks at the semantic level enhances content traceability against strong paraphrasing attacks, providing a reliable tool for AI governance. From an ethical perspective, watermarking techniques could be misused to track individual writing styles, raising potential privacy concerns. It is therefore critical to establish responsible deployment guidelines that balance safety auditing with the protection of user anonymity to foster a transparent and trustworthy ecosystem for generative artificial intelligence.

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

# Appendix

## A. Additional Experimental Results

**Impact of Generated Text Length on Detection Performance.** The reliability of watermark detection fundamentally depends on the volume of statistical information available within an observed sequence. The evaluation of PASA across varying text lengths reveals a high degree of efficiency in low-resource scenarios. At a sequence length of only 50 tokens, the ROC-AUC already exceeds 0.95, indicating that shared randomness anchored at the semantic level provides significant discriminative power even within a minimal context. As the length extends to 300 tokens, stringent metrics such as the TPR@1%FPR rapidly converge toward 1.0, a trend characterized in Figure 4. This efficient convergence is attributed to the stability of semantic clusters, which mitigates the impact of local token-level noise and allows the watermark signal to reach statistical significance within short durations. These findings confirm that PASA is highly practical for real-world applications involving short-form content or latency-sensitive generations.

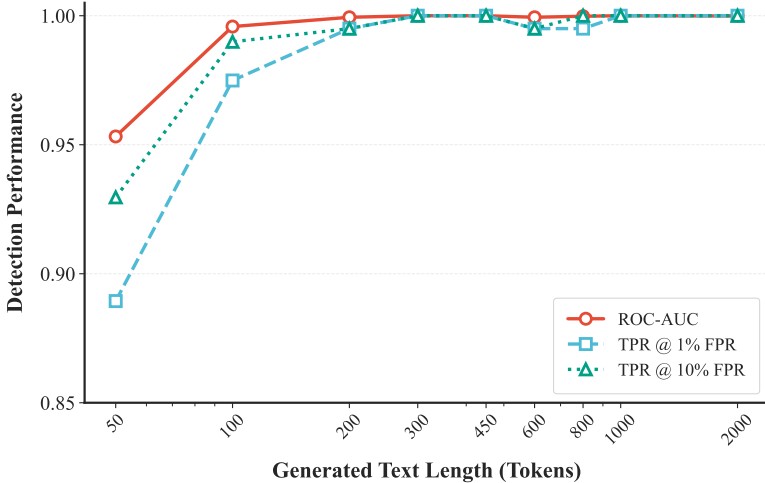

*Figure 4.* **Detection performance across various generated text lengths.** The ROC-AUC and True Positive Rate (TPR) exhibit rapid convergence, achieving near-perfect detection beyond 300 tokens.

**Generalization Analysis on the ELI5 Dataset.** The ELI5 dataset is designed for long-form question answering, requiring models to produce detailed explanations for complex queries. We use this dataset to evaluate the generalization of PASA in linguistic contexts beyond standard news benchmarks. As shown in Table 4, PASA maintains near-perfect detection accuracy on clean text, matching state-of-the-art baselines. Under the T5-Large token-level replacement attack, PASA exhibits superior robustness, achieving the highest ROC-AUC (0.9980) and TPR@1%FPR (0.9750). In contrast, methods such as DAWA degrade substantially under attack, whereas PASA effectively mitigates the state mismatch induced by local edits. These results suggest that anchoring shared randomness in the latent semantic space yields stable watermark survivability across generation tasks and data distributions, supporting the broad applicability of our framework.

*Table 4.* **Detection performance on ELI5 dataset under token-replacement attacks.** Comparisons of ROC-AUC, TPR@1%FPR, and TPR@10%FPR using the LLAMA-13B-hf architecture. For token-replacement attacks with replacement ratio $r = 0.5$, we use T5-Large as the attacker. **Best**, Second Best, and Third Best results are marked in each column.

| LLM | Method | Clean Text (ELI5) | | | T5-Large Attack | | |
|---|---|---|---|---|---|---|---|
| | | ROC-AUC↑ | TPR@1%FPR↑ | TPR@10%FPR↑ | ROC-AUC↑ | TPR@1%FPR↑ | TPR@10%FPR↑ |
| LLAMA-13B | DAWA | **1.0000** | **1.0000** | **1.0000** | 0.8784 | 0.5150 | 0.6900 |
| | Exp-edit | 0.9908 | 0.9900 | 0.9900 | 0.9737 | 0.9650 | 0.9650 |
| | AWTI | **1.0000** | **1.0000** | **1.0000** | 0.9975 | 0.9550 | **1.0000** |
| | Ours | **1.0000** | **1.0000** | **1.0000** | **0.9980** | **0.9750** | 0.9950 |

*Table 5.* **Additional robustness comparison under diverse paraphrasing attacks.** We compare ROC-AUC, TPR@1%FPR, and TPR@10%FPR under the clean setting and three semantic-invariant attacks, including DIPPER, OPT-2.7B paraphrasing, and WM-removal. **Best**, Second Best, and Third Best results are marked when applicable.

| Attack setting | Metric | AWTI | SIR | SynthID-Text | Ours |
|---|---|---|---|---|---|
| No attack | ROC-AUC↑ | **1.0000** | **1.0000** | 0.9871 | **1.0000** |
| | TPR@1%FPR↑ | **1.0000** | **1.0000** | 0.9400 | **1.0000** |
| | TPR@10%FPR↑ | **1.0000** | **1.0000** | 0.9900 | **1.0000** |
| DIPPER | ROC-AUC↑ | 0.7391 | 0.8809 | 0.6228 | **0.8956** |
| | TPR@1%FPR↑ | 0.1350 | 0.2900 | 0.0200 | **0.5377** |
| | TPR@10%FPR↑ | 0.3600 | **0.8809** | 0.2700 | 0.7593 |
| OPT-2.7B | ROC-AUC↑ | 0.9659 | 0.9623 | 0.9483 | **0.9931** |
| | TPR@1%FPR↑ | 0.8400 | 0.5950 | 0.4350 | **0.9146** |
| | TPR@10%FPR↑ | 0.9150 | 0.9350 | 0.8650 | **0.9749** |
| WM-removal | ROC-AUC↑ | 0.9635 | 0.9908 | 0.9772 | **0.9972** |
| | TPR@1%FPR↑ | 0.7050 | 0.8550 | 0.6500 | **0.9598** |
| | TPR@10%FPR↑ | 0.8450 | **0.9850** | 0.9600 | 0.9849 |

**Additional Comparison under Diverse Paraphrasing Attacks.** To further evaluate the robustness of PASA, we extend the comparison to additional watermarking baselines and stronger paraphrasing attacks. Specifically, we include SIR (Liu et al., 2024b), a representative semantic watermarking method, and SynthID-Text (Dathathri et al., 2024), a representative distortion-free watermarking scheme. We evaluate all methods under three paraphrasing-based attacks, including DIPPER, OPT-2.7B paraphrasing (Zhang et al., 2022), and WM-removal. Following the main experiments, we report ROC-AUC, TPR@1%FPR, and TPR@10%FPR.

Table 5 presents the results. Under the no-attack setting, PASA achieves perfect detection performance, matching the strongest baselines. Under semantic-invariant attacks, PASA consistently achieves the highest ROC-AUC and TPR@1%FPR across all attack settings, demonstrating stronger robustness in the stricter low-false-positive regime. In particular, under OPT-2.7B paraphrasing and WM-removal, PASA achieves ROC-AUC scores of 0.9931 and 0.9972, with TPR@1%FPR of 0.9146 and 0.9598, respectively. These results indicate that PASA preserves a more stable detection signal under diverse meaning-preserving transformations. Although SIR obtains a higher TPR@10%FPR under DIPPER and a marginally higher TPR@10%FPR under WM-removal, PASA provides consistently stronger performance at TPR@1%FPR, which is more critical for reliable watermark detection in practical low-FPR scenarios.

**Detection with and without Prompts.** Since prompts play a central role in conditioning LLM generation (Feng et al., 2025; Zhu et al., 2025a; Jin et al., 2025) and can substantially affect model behavior and evaluation outcomes (Tao et al., 2026; Zhang et al., 2025b), we further clarify the detection setting used in our experiments. Unless otherwise stated, detection is performed only on the generated continuation, without including the input prompt, for both C4 and ELI5. This setting avoids introducing prompt-specific artifacts into the detector and ensures that the reported performance reflects the detectability of the watermarked generation itself.

We additionally evaluate a more conservative setting where the human-written prompt is prepended to the watermarked continuation before detection. This setting simulates a simple mixed-text scenario in which unwatermarked human text appears before the watermarked passage, thereby diluting the watermark signal available to the detector. As shown in Table 6, PASA remains reliably detectable even when the prompt is included. On C4, PASA achieves an ROC-AUC of 0.9997 and a TPR@1%FPR of 0.9899, while maintaining a TPR@10%FPR of 1.0000. On ELI5, detection performance remains perfect under both settings. These results indicate that PASA is robust to prompt prepending and that its detection signal primarily comes from the generated continuation rather than prompt-specific artifacts.

**Robustness under Surrogate LM Mismatch.** We further evaluate the robustness of PASA under surrogate language model (SLM) mismatch. Specifically, we consider both detector-side SLM mismatch and base/instruction-tuned mismatch within the LLAMA-2 family. No per-distribution calibration is used in this experiment.

*Table 6.* **Detection performance with and without prompts.** We compare ROC-AUC, TPR@1%FPR, and TPR@10%FPR on C4 and ELI5. "Without prompt" denotes detection on the generated continuation only, while "Mixed with prompt" denotes detection after prepending the human-written prompt to the watermarked continuation.

| Setting | C4 | | | ELI5 | | |
|---|---|---|---|---|---|---|
| | ROC-AUC↑ | TPR@1%FPR↑ | TPR@10%FPR↑ | ROC-AUC↑ | TPR@1%FPR↑ | TPR@10%FPR↑ |
| Without prompt | **1.0000** | **1.0000** | **1.0000** | **1.0000** | **1.0000** | **1.0000** |
| Mixed with prompt | 0.9997 | 0.9899 | **1.0000** | **1.0000** | **1.0000** | **1.0000** |

*Table 7.* **Detection performance under surrogate LM mismatch.** We evaluate detector-side SLM mismatch and base/instruction-tuned mismatch within the LLAMA-2 family. ROC-AUC, TPR@1%FPR, and TPR@10%FPR are reported. No per-distribution calibration is used.

| Generation Model | Detector-side SLM | ROC-AUC↑ | TPR@1%FPR↑ | TPR@10%FPR↑ |
|---|---|---|---|---|
| LLAMA-2-13B | LLAMA-2-7B | **1.0000** | **1.0000** | **1.0000** |
| LLAMA-2-13B | TinyLLAMA-1.1B | **1.0000** | **1.0000** | **1.0000** |
| LLAMA-2-13B | LLAMA-2-7B-chat | 0.9995 | 0.9950 | 0.9950 |
| LLAMA-2-13B-chat | LLAMA-2-7B | 0.9817 | 0.8750 | 0.9500 |
| LLAMA-2-13B-chat | LLAMA-2-7B-chat | 0.9879 | 0.8800 | 0.9800 |

As shown in Table 7, PASA remains highly reliable when the detector-side SLM comes from the same model family or uses a compatible tokenizer. When the generation model is LLAMA-2-13B, both LLAMA-2-7B and TinyLLAMA-1.1B (Zhang et al., 2024a) achieve perfect detection performance, with ROC-AUC, TPR@1%FPR, and TPR@10%FPR all reaching 1.0000. This suggests that preserving tokenizer compatibility and the next-token prediction structure is more important than the scale of the detector-side SLM. Even when using LLAMA-2-7B-chat as the detector-side SLM, PASA still achieves a ROC-AUC of 0.9995 and a TPR@1%FPR of 0.9950.

When the generation model is instruction-tuned, detection performance slightly decreases but remains reliable. For example, when detecting text generated by LLAMA-2-13B-chat, PASA achieves ROC-AUC scores of 0.9817 and 0.9879 with LLAMA-2-7B and LLAMA-2-7B-chat as detector-side SLMs, respectively. A possible explanation is that instruction tuning changes the next-token distribution and makes generations more structured and constrained, thereby reducing the effective randomness available to the statistical detector.

Overall, these results suggest that PASA is most effective within the same model family or tokenizer-compatible model families. From a deployment perspective, a practical solution is to maintain several lightweight local detectors corresponding to different candidate model families. A high-confidence detection signal from one detector can both verify the watermark and provide evidence about the likely source LLM family.

**Computational and Memory Costs.** We further report the memory requirements and computational costs of different watermarking methods during generation and detection. For generation, all methods use the same generation backbone under our setup, which requires 25,376 MB of GPU memory. Therefore, the generation-side memory requirement is essentially identical across methods, and the main difference lies in the detection stage.

As shown in Table 8, PASA maintains moderate detection-side cost compared with existing watermarking methods. In terms of detection memory, PASA requires 2,892 MB, which is comparable to DAWA and lower than AWTI. In terms of computational cost, PASA requires $6.21 \times 10^2$ GFLOPs, which is lower than DAWA, SIR, and AWTI. These results show that PASA achieves strong robustness while maintaining practical detection-side efficiency.

# B. Related Works

**LLM Text Watermarking.** Prior surveys of watermarking for Large Language Models (LLMs) (Liu et al., 2024c; Yang et al., 2025b) typically categorize methods by whether the watermark is embedded pre-generation or post-generation, and they organize evaluations along dimensions such as detectability, impact on text quality, robustness, and security. Within this taxonomy, in-generation watermarking has emerged as a dominant paradigm due to its direct integration into the decoding and sampling processes, thereby incurring minimal overhead during deployment.

*Table 8.* **Memory requirements and computational costs.** All methods use the same generation backbone, which requires 25,376 MB of GPU memory under our setup. We report detection-side memory usage and GFLOPs.

| Metric | DAWA | EXP-edit | AWTI | SIR | Ours |
|---|---|---|---|---|---|
| Detection Memory (MB) | 2892 | CPU | 6344 | 1994 | 2892 |
| Detection GFLOPs | $1.06 \times 10^3$ | $5.12 \times 10^{-5}$ | $2.85 \times 10^5$ | $1.32 \times 10^4$ | $6.21 \times 10^2$ |

For watermark generation, the classic green-list approach (Kirchenbauer et al., 2023) uses a secret-key-driven partition of the vocabulary to induce a slight sampling bias toward green tokens, and subsequently applies interpretable statistical tests to compute detection p-values. Another line of work pursues distortion-free (distribution-preserving) embedding by incorporating detectable signals while maintaining the original generation distribution of the model, either implicitly or explicitly. Representative methods align randomness derived from a secret key with the sampling procedure of the language model, enabling detection by re-synchronizing and validating the induced shared randomness. For instance, the Gumbel-Max watermark achieves exact sampling of the next token via the Gumbel–Max trick (Aaronson, 2023; Gumbel, 1954), whereas an inverse transform construction provides an alternative instantiation of exact sampling (Kuditipudi et al., 2024).

Production-oriented watermarking systems have advanced rapidly in recent years. For example, SynthID Text (Dathathri et al., 2024) targets production readiness by modifying only the sampling procedure (without retraining) while maintaining efficient detection and low-latency overhead. Concurrently, prior work has addressed the degradation of diversity induced by decoding-based watermarks, proposing Gumbel–Max variants (Fu et al., 2024a) that better balance generative diversity and detectability. Furthermore, investigations into the learnability of watermarks (Gu et al., 2024) indicate that models are capable of distilling watermarking behavior, thereby enabling the generation of watermarked text. While this phenomenon supports watermarking in open-source environments, it simultaneously elevates the risk of watermark forgery by adversaries, which could facilitate attribution attacks.

Methodologically, in contrast to token-based approaches, we elevate both the embedding and verification units from individual tokens to semantic clusters in the embedding space. By constructing detection statistics from cluster-level shared randomness, our design directly targets robustness to semantics-preserving rewriting perturbations.

**Theoretical Works.** Beyond empirical heuristics, a growing body of work characterizes watermark detectability, quality, and robustness through statistical testing and formal analysis. The foundational Green-list approach (Kirchenbauer et al., 2023) not only introduced detection statistics and associated p-values, but also analyzed how detection sensitivity varies with generation uncertainty, establishing a widely used analytical baseline. Building on this line, Unigram Watermark (Zhao et al., 2024) proposed a rigorous framework for quantifying validity and robustness, providing provable guarantees under perturbations such as random edits and paraphrasing. More recently, DAWA (He et al., 2025) has emphasized the construction of distribution-adaptive and distortion-free schemes motivated by theoretical optimality. By leveraging surrogate models to enable model-agnostic detection, DAWA has demonstrated robust performance, particularly within the regime of ultra-low false positive rates. Theoretically, our work extends this framework by specifically modeling semantic-invariant attacks to incorporate robustness into design.

**Robustness and Attacks.** Robustness remains a critical bottleneck for the real-world deployment of text watermarking. Attacks that preserve semantics, such as controlled token replacement and paraphrasing, can substantially alter surface token sequences, causing rapid signal decay in methods that treat individual tokens as the fundamental unit. While studies on distortion-free watermarking (Kuditipudi et al., 2024) suggest resilience to random edits (substitutions, insertions, deletions) and mild automated rewriting, they also indicate that low-entropy generation or aggressive paraphrasing can severely compromise detection efficacy. From the perspective of trade-offs, WaterMax (Giboulot & Furon, 2024) targets a joint balance among detectability, robustness, and quality, showing that strong performance can be achieved without modifying model weights or sampling mechanisms. More recently, SEEK (Shen et al., 2025) identified a trade-off between scrubbing and spoofing attacks driven by window size, proposing "equivalent texture keys" and redundancy mechanisms to strengthen defenses against both threats. Closely related to robustness is the risk of forgery: studies concerning the learnability of watermarks (Gu et al., 2024) suggest that adversaries can train models to generate text that detectors accept as watermarked, posing a significant spoofing threat. For evaluation, we employ strong semantic-preserving attacks, including T5-based replacement (Raffel et al., 2020) and DIPPER paraphrasing (Krishna et al., 2023), focusing on detection performance under these attacks at low false positive rates (FPR).

## C. Proof of Theorem 1

According to the worst-case FA error constraint, we have $\forall x^T \in \mathcal{V}^T$,

$$
\alpha \geq \max_{Q_{X^T}} \mathbb{E}_{Q_{X^T} \otimes P_{\zeta^T}} \left[ \sup_{\tilde{x}^T \in \mathcal{B}_f(X^T)} \mathbb{1}\{\gamma(\tilde{x}^T, \zeta^T) = 1\} \right]
$$

$$
\geq \mathbb{E}_{\delta_{x^T} \otimes P_{\zeta^T}} \left[ \sup_{\tilde{x}^T \in \mathcal{B}_f(X^T)} \mathbb{1}\{\gamma(\tilde{x}^T, \zeta^T) = 1\} \right] = \mathbb{E}_{P_{\zeta^T}} \left[ \sup_{\tilde{x}^T \in \mathcal{B}_f(x^T)} \gamma(\tilde{x}^T, \zeta^T) \right]
$$

$$
= \sum_{\zeta^T} P_{\zeta^T}(\zeta^T) \sup_{\tilde{x}^T \in \mathcal{B}_f(x^T)} \gamma(\tilde{x}^T, \zeta^T).
$$

For brevity, let $\mathcal{B}(k) := \mathcal{B}_f(x^T)$ if $f(x^T) = k$. The MD error is equal to $1 - \mathbb{E}_{P_{X^T,\zeta^T}}[\inf_{\tilde{x}^T \in \mathcal{B}_f(X^T)} \gamma(\tilde{x}^T, \zeta^T)]$. Thus, to lower bound the MD error, we first upper bound the second term

$$
\mathbb{E}_{P_{X^T,\zeta^T}} \left[ \inf_{\tilde{x}^T \in \mathcal{B}_f(X^T)} \gamma(\tilde{x}^T, \zeta^T) \right] \leq \mathbb{E}_{P_{X^T,\zeta^T}} \left[ \sup_{\tilde{x}^T \in \mathcal{B}_f(X^T)} \gamma(\tilde{x}^T, \zeta^T) \right]
$$

$$
= \sum_{k \in [K]} \underbrace{\sum_{x^T : f(x^T) = k} \sum_{\zeta^T} P_{X^T,\zeta^T}(x^T, \zeta^T) \sup_{\tilde{x}^T \in \mathcal{B}_f(x^T)} \gamma(\tilde{x}^T, \zeta^T)}_{C(k)},
$$

where according to the FA error constraint, for all $k \in [K]$,

$$
C(k) \leq \sum_{x^T : f(x^T) = k} P_{X^T}(x^T), \quad \text{and}
$$

$$
C(k) = \sum_{\zeta^T} P_{\zeta^T}(\zeta^T) \sum_{x^T : f(x^T) = k} P_{X^T|\zeta^T}(x^T|\zeta^T) \sup_{\tilde{x}^T \in \mathcal{B}(k)} \gamma(\tilde{x}^T, \zeta^T)
$$

$$
\leq \sum_{\zeta^T} P_{\zeta^T}(\zeta^T) \sup_{\tilde{x}^T \in \mathcal{B}(k)} \gamma(\tilde{x}^T, \zeta^T) \leq \alpha.
$$

Therefore,

$$
\mathbb{E}_{P_{X^T,\zeta^T}} \left[ \inf_{\tilde{x}^T \in \mathcal{B}(f(X^T))} \gamma(\tilde{x}^T, \zeta^T) \right] \leq \sum_{k \in [K]} C(k)
$$

$$
\leq \sum_{k \in [K]} \left( \left( \sum_{x^T : f(x^T) = k} P_{X^T}(x^T) \right) \wedge \alpha \right) = 1 - \sum_{k \in [K]} \left( \left( \sum_{x^T : f(x^T) = k} P_{X^T}(x^T) \right) - \alpha \right)_+, \tag{2}
$$

where (2) is maximized by taking

$$
P_{X^T} = P_{X^T}^* := \operatorname*{arg\,min}_{P_{X^T} : \mathsf{D}(P_{X^T}, Q_{X^T}) \leq \epsilon} \sum_{k \in [K]} \left( \left( \sum_{x^T : f(x^T) = k} P_{X^T}(x^T) \right) - \alpha \right)_+.
$$

Finally, the MD error is lower bounded by

$$
\beta_1^f(\gamma, P_{X^T,\zeta^T}) = 1 - \mathbb{E}_{P_{X^T,\zeta^T}} \left[ \inf_{\tilde{x}^T \in \mathcal{B}_f(X^T)} \gamma(\tilde{x}^T, \zeta^T) \right]
$$

$$
\geq \sum_{k \in [K]} \left( \left( \sum_{x^T : f(x^T) = k} P_{X^T}^*(x^T) \right) - \alpha \right)_+
$$

$$
= \min_{P_{X^T} : \mathsf{D}(P_{X^T}, Q_{X^T}) \leq \epsilon} \sum_{k \in [K]} \left( \left( \sum_{x^T : f(x^T) = k} P_{X^T}(x^T) \right) - \alpha \right)_+.
$$

In the next section, we prove that there exists a watermark embedding-detection pair that achieves this lower bound. Therefore, this lower bound is the optimal objective value of the optimization problem (P):

$$
\beta_1^{f,*} := \min_{P_{X^T} : \mathsf{D}(P_{X^T}, Q_{X^T}) \leq \epsilon} \sum_{k \in [K]} \left( \left( \sum_{x^T : f(x^T) = k} P_{X^T}(x^T) \right) - \alpha \right)_+.
$$

When $\epsilon = 0$, it becomes the minimum MD error for a distortion-free watermarking scheme:

$$
\beta_1^{f,*}(\epsilon = 0) := \sum_{k \in [K]} \left( \left( \sum_{x^T : f(x^T) = k} Q_{X^T}(x^T) \right) - \alpha \right)_+.
$$

## D. Proof of Theorem 2

**Theorem 2** ((Formal) Jointly Optimal Watermark Embedding and Detection under $f$ Attack)**.** *Let $\Gamma_f^*$ be a collection of detectors that accept the form*

$$\gamma(X^T, \zeta^T) = \mathbb{1}\{f(X^T) = \text{vec2num}(\zeta^T)\}$$

*where $\text{vec2num} : \mathcal{Z}^T \to [K] \cup \{\tilde{\zeta}\}$ is a bijective function that maps a sequence to a real number and $\tilde{\zeta} \in \mathbb{N} \setminus [K]$ is called the overflow state.*

*For any detector $\gamma \in \Gamma_f^*$, the corresponding distortion-free and robust watermark embedding method $P_{X^T, \zeta^T}^*$ that together achieves the minimum MD error attained from ([P](#)) reaches $\beta_1^{f,*}$ in Theorem 1 is given as follows:*

1. *the auxiliary sequence distribution $P_{\zeta^T}$:*

$$\begin{cases} P_{\zeta^T}^*(\zeta^T) = \sum_{k \in [K]} \mathbb{1}\{k = \text{vec2num}(\zeta^T)\}\left(\sum_{x^T \in \mathcal{V}^T : f(x^T) = k} Q_{X^T}(x^T)\right) \wedge \alpha, & \forall \zeta^T \text{s.t. } \text{vec2num}(\zeta^T) \neq \tilde{\zeta}, \\ P_{\zeta^T}^*(\zeta^T) = \sum_{k \in [K]} \left(\sum_{x^T : f(x^T) = k} Q_{X^T}(x^T) - \alpha\right)_+, & \text{if } \text{vec2num}(\zeta^T) = \tilde{\zeta}; \end{cases}$$

2. *the conditional token sequence distribution $P_{X^T|\zeta^T}^*$: for any $x^T \in \mathcal{X}^T$,*

$$\begin{cases} P_{X^T|\zeta^T}^*(x^T|\zeta^T) = \frac{\mathbb{1}\{f(x^T) = \text{vec2num}(\zeta^T)\}Q_{X^T}(x^T)}{\sum_{x^T : f(T) = k} Q_{X^T}(x^T)}, & \forall \zeta^T \text{s.t. } \text{vec2num}(\zeta^T) \neq \tilde{\zeta}, \\ P_{X^T|\zeta^T}^*(x^T|\zeta^T) = \frac{Q_{X^T}(x^T)}{\sum_{v^T : f(v^T) = f(x^T)} Q_{X^T}(v^T)} \frac{(\sum_{v^T : f(v^T) = f(x^T)} Q_{X^T}(v^T) - \alpha)_+}{\sum_{k \in [K]}(\sum_{v^T : f(v^T) = k} Q_{X^T}(v^T) - \alpha)_+}, & \text{if } \text{vec2num}(\zeta^T) = \tilde{\zeta}. \end{cases}$$

*Proof.* Under a detector $\gamma \in \Gamma_f^*$ and the corresponding watermarking method $P_{X^T, \zeta^T}^*$, the induced MD and worst-case FA errors are given by:

**Worst-case FA error:**

$$\because \forall y_1^T \in \mathcal{V}^T, \quad \mathbb{E}_{P_{\zeta^T}^*}\left[\sup_{\tilde{x}^T \in \mathcal{B}_f(y_1^T)} \mathbb{1}\{\gamma(\tilde{x}^T, \zeta^T) = 1\}\right]$$

$$= \sum_{\zeta^T} P_{\zeta^T}^*(\zeta^T) \sup_{\tilde{x}^T \in \mathcal{B}_f(y_1^T))} \mathbb{1}\{\gamma(\tilde{x}^T, \zeta^T) = 1\}$$

$$= \left(\sum_{x^T \in \mathcal{V}^T : f(x^T) = f(y^T)} Q_{X^T}(x^T)\right) \wedge \alpha \leq \alpha$$

and since any distribution $Q_{X^T}$ can be written as a linear combinations of $\delta_{y_1^T}$,

$$\therefore \sup_{Q_{X^T}} \mathbb{E}_{Q_{X^T} \otimes P_{\zeta^T}^*}\left[\sup_{\tilde{x}^T \in \mathcal{B}_f(X^T)} \mathbb{1}\{\gamma(\tilde{x}^T, \zeta^T) = 1\}\right] \leq \alpha.$$

**MD error:**

$$\mathbb{E}_{P_{X^T, \zeta^T}^*}\left[\sup_{\tilde{x}^T \in \mathcal{B}_f(X^T)} \mathbb{1}\{\gamma(\tilde{x}^T, \zeta^T) = 0\}\right]$$

$$= \sum_{x^T} P_{X^T, \zeta^T}^*(x^T, \text{vec2num}^{-1}(\tilde{\zeta})) + \underbrace{\sum_{k \in [K]} \sum_{x^T : f(x^T) = k} \sum_{\zeta^T : \text{vec2num}(\zeta^T) \neq \tilde{\zeta}} P_{X^T, \zeta^T}^*(x^T, \zeta^T) \sup_{\tilde{x}^T \in \mathcal{B}(k)} \mathbb{1}\{\gamma(\tilde{x}^T, \zeta^T) = 0\}}_{=0}$$

$$= \sum_{k \in [K]} \left(\left(\sum_{x^T \in \mathcal{B}(k)} Q_{X^T}(x^T)\right) - \alpha\right)_+ = \beta_1^{f,*}.$$

The optimality is thus proved. $\qquad\square$

## E. Examples of Watermarked Text

In this section, we provide representative examples of watermarked text generated by PASA, as shown in Table 9. Each example consists of an input prompt and its corresponding watermarked continuation. These examples cover diverse domains, including wildlife conservation, public policy, technology, history, entertainment, travel, and news reporting. They

*Table 9.* Examples of watermarked text generated by PASA across diverse domains.

| Domain | Prompt | Watermarked Text |
|---|---|---|
| Wildlife Conservation | **Example 1.** Back in 1982, the Endangered Species Act took the ocelot under its protection. The ocelot is a vulnerable creature, susceptible to habitat changes like roads, agriculture, housing developments and trapping. | Recently, Wild Earth Guardians made another effort to increase the protection of the ocelot. The Kenedy Ranch, a Texas ranch, located in the Lower Rio Grande Valley, was set to build a huge housing development that would have been fatal to the ocelots who made their home there. When Wild Earth Guardians protested, the Kenedy Ranch switched over to renting out cabin spaces for money instead. The buildings Ocelots like to live in would now be used by human construction contractors as dormitories during the construction process. This change of heart benefited both parties and gave the wildlife a fighting chance. Even though it is a small step, the additional protection of the ocelot definitely starts the improvement of endangered species. |
| Public Policy | **Example 2.** On Jan. 15, nine of 12 members of the National Park System Advisory Board sent a letter to Interior Secretary Ryan Zinke announcing their resignation, effectively dissolving the board. It was an act of protest against an administration with little appetite for the methodical approach the board has brought to national park management for decades. | In the 100-plus-year history of the National Park Service, the board is an innovation — an organization that provides a forum for professionals not from park schools, and that is not beholden to the whim of the NPS, the inmates, as it were, governing the asylum. In an age when bureaucracies seem beholden to the mood of the mob, the board's mission to resist ideology and opinion and secure methodological governance is no small thing. To keep the parks from devolving to the state of mood and opinion, the board's work is extraordinarily important. But the NPS — and the board that has served it so well — faces precarious challenges. A perennially at-risk budget, climate changes, shifting demographics and a lack of trained personnel are demanding resistance on the board's part. |
| Technology | **Example 3.** It's amazing what a difference time can make in the way we perceive certain technologies. A few years ago, the idea of the cloud was terrifying to many people. | It was a foreign concept to say the least. And open source software was something used by people in dangerous areas of the world (or at school) so it wasn't exactly a "threat" to the .com bubble in the early 2000's, but it wasn't exactly the shiniest of jewels in the IT world. These terms hovered around the peripheries of IT until recently, and now they're front and center. Though the stigmas have gone away, that doesn't mean the underlying realities have changed. In fact, over time the way we've looked at cloud computing has changed tremendously, and a lot of that has to do with open source software (which is not to be confused with part of the cloud). |
| History | **Example 4.** Readers may be familiar with the fundamental changes that took place in the Roman world as it converted from paganism to Christianity in the fourth century, and as its emperors sought to govern, through the turbulent times of the fifth to seventh centuries, as Christian rulers. This is the stuff of late antiquity as it would be recognised in any classics or history university department. | For classical archaeologists, the fifth century is a time when the writing on tombstones disappeared (at least partly because the names, place names, and details of public offices on the tombstones were all interpreted as carrying pagan cues in a church where it was assumed that everyone was a Christian), when tombs could be massively constructed and decorated with biblical or lengthy Christian inscriptions (in contrast to the short, formulaic pagan inscriptions), and when gravestones bore the Latin cross, or a gibbet held by two feet, a symbol of martyrdom. A rhetorically 'Origenean' fifth century fountain-of-youth discourse is emblematic of the spiritual and physical pain of the human body, and the re-shaping of landscape around the Christian cult of relics. |
| Entertainment | **Example 5.** A large, brand-new movie studio in Palm Beach County already has a potential box-office hit. Twentieth Century Fox will film interior scenes for Speed II, sequel to the popular 1994 action movie, in the 20,000-square-foot sound stage at Palm Beach Ocean Studios in West Palm Beach, the studio's chief executive said on Thursday. | The location will substitute for the Chicago interior of the movie's headquarters of the fictitious Inner City Transportation System, said Robert J. Stein, chief executive of the studio at the former South Florida Video Arts Institute, 2721 Vista Parkway. The movie, due out in 1997, is the first big-budget film production attracted to the 10-month-old studio, which has beefed up its facilities since opening in May and offers one of the most modern designs of any studio in the country. "It wasn't a very exciting utilization of the studio up to now," Stein said. "And then it's a very exciting utilization." The studio also hopes to land more projects in the future. It has lined up a call sheet, or schedule of local props and actors to be used if additional portions of the movie are filmed in the area. |
| Travel | **Example 6.** As the crow flies Denniff's Cottage on Ringarogey island in Roaring Water Bay is just a few minutes' boat ride to Baltimore, Co Cork. In contrast, the scenic route is by car, a journey time around the winding roads, of about 20 minutes. | Either way, Denniff's Cottage is a perfect stopover base for a few nights, and with just a 10-15 minute walk the convenience and peaceful setting of its location on the island, truly make it a gem indeed. The cottage is in an enviable situation with direct access to the seafront and a small beach, and the en-suite bedrooms have simply sensational sea views. The stylish property is well furnished and decorated, and the spacious open plan kitchen/dining/living area is perfect for relaxing and socialising. There is a stove for cosy evenings, and outside there are a variety of places to sit and watch the sun go down over Baltimore. Denniff's Cottage is just a 10-15 minute walk from the village of Baltimore. |
| News Reporting | **Example 7.** BAGHDAD, Iraq – Fighting in the Shiite stronghold of Sadr City killed 23 Iraqis yesterday, hospital officials said, and the U.S. military reported five troop deaths, as April showed signs of becoming the worst month for U.S. forces in Iraq since September. At least 11 of the Iraqi deaths occurred when mortar shells landed in residential neighborhoods. | Two Iraqi police commanders and about 100 of their officers deserted their posts across the western city of Fallujah early yesterday, police said. The police said some of the deserting officers were in the past day or so arrested by American troops in a net tightening around Fallujah. The U.S. military has been blocking access to the city amidst increasing reports of fighting. Black smoke rose from a market area in the southern city of Hilla, and Fallujah residents said at least one neighborhood there was on fire. Fallujah hospitals reported six deaths and 100 wounded over the past day, an official at Fallujah General Hospital said. In Baghdad, gunmen in a car fatally shot Abu Kawthar, a Shiite Muslim cleric, along with his guard, in the Jihad neighborhood, police said. |

demonstrate that PASA can generate natural, fluent, and semantically coherent continuations while maintaining high text quality and successfully embedding watermark signals.

# F. Implementation Details.

**Hyperparameters.**   During watermarked text generation, we employ multinomial sampling (`top_p`=1.0) with a fixed temperature $\tau = 1.0$. The watermark embedding process initiates after the first three precursor tokens. For the specific hyperparameters of PASA, we set the semantic cluster number to $K = 4$, the synchronization window size to $w = 3$, and the FA threshold to $\alpha = 0.4$. Unless otherwise stated, the length of generated text is constrained to the range of 200 to 300 tokens. These configurations were empirically selected via ablation studies to optimize the trade-off between robustness and detectability. All experiments were conducted using a single NVIDIA RTX PRO 6000 GPU.

**DIPPER Attack Configurations.**   To evaluate robustness against sophisticated paraphrasing, we utilize the DIPPER model with three escalating intensities. These are defined by varying the Lexical Diversity (L) and Word Order Diversity (O) parameters as follows:

- **Level 1 (Lexical Substitution)**: (L=60,O=0), focusing on heavy synonymous replacement.
- **Level 2 (Moderate Reordering)**: (L=60,O=20), combining lexical changes with moderate structural shifts.
- **Level 3 (Syntactic Restructuring)**: (L=60,O=80), representing aggressive syntactic modifications.

We evaluate PASA across a broad spectrum of semantic-invariant attacks, including DIPPER paraphrasing. Beyond standard configurations, we test escalating intensities such as (L=60, O=40) and (L=60, O=60) to characterize performance under aggressive syntactic and lexical restructuring.

