# OpenReview forum: "PASA: A Principled Embedding-Space Watermarking Approach for LLM-Generated Text under Semantic-Invariant Attacks"
_ICML.cc/2026/Conference — ICML 2026 regular_

### Official Review · Reviewer_oVUC · 2026-02-15

**Soundness:** 3
**Presentation:** 3
**Significance:** 3
**Originality:** 2
**Overall Recommendation:** 4
**Confidence:** 4

**Summary:**

Due to the vulnerability of proposed text watermarks to semantic-invariant attacks, the authors proposed a new method which takes watermarking from the token space to a more semantic space (latent) by embedding and detecting watermarks through a distributional dependency between token sequences and auxiliary random sequences. They provided a theoretical framework that characterizes a jointly optimal embedding-detection pair to achieve a good trade-off between detection accuracy, robustness, and distortion. Empirical results show that their proposed approach remains robust even under strong paraphrasing attack while preserving quality, in comparison to standard token-space watermarks.

**Compliance With Llm Reviewing Policy:**

Affirmed.

**Final Justification:**

The authors fully addressed my concerns and I am convinced by the method and the newly presented results. The only drawback is the originality but my confidence in the necessity of the approach as well as significance would make me lean for an accept. Furthermore, I would be expecting the authors to add qualitative examples, failure nodes, and include the new results in the final version.

**Key Questions For Authors:**

1. Just like I stated above, answer these questions with evidence:
(a) is this approach better than current semantic watermarks?
(b) is it faster or more compute intense?
(c) does it bring any further advantage to robustness over other proposed semantic watermarks?

2. What are the failure modes? What caused the detector to fail? What are potential future works to stop this from happening?

3. Can you present some qualitative results in the appendix? If possible, can you make changes to Figure 2 to make it more intuitive?

**Limitations:**

Yes

**Strengths And Weaknesses:**

**Strength**: The authors proposed a new method inspired by classic token-distribution based watermarks, but takes that idea to the semantic space to create something more robust. They rolled out a theoretical framework backing their approach, and further supplemented it with empirical evidence showing that they achieve a desirable tradeoff between distortion, quality, and detectability. I also appreciate the unique writing style to best pass the message they are trying to send, as well as how the sections are organized. Their focus against semantic invariant attacks is well-specified.

**Weakness**: Figure 2 (probably the most important one in the paper) is not clear enough. I understood their approach reading through their methodology and caption than the image itself. This is a minor one, as Figure 1 already shows a teaser for anyone to understand the high-level. However, it would be nice if Figure 2 can be refined and improved. **Another** issue comes with the surveyed baselines, which are all token-space watermarks. Current semantic watermarks should be surveyed too. Questions like (i) is this approach better than current semantic watermarks? (ii) is it faster or more compute intense? (iii) what exact advantage do we bring to robustness over other proposed semantic watermarks?, should be answered. **Finally**, failure modes were not properly discussed, even with empirical trade-off evidence being presented. When detection fails, what had caused it? How do we prevent such from happening? Provide some qualitative examples in the appendix too so that the work won't look too heavy on theorems and numbers, but with some real evidence showcasing catches and misses.

---

> ### Author Rebuttal · Authors · 2026-03-31
>
> Response to Reviewer `oVUC`:
>
> Thank you so much for your constructive and insightful comments! We address your concerns point by point as follows.
>
>
> `Q1:` Just like I stated above, answer these questions with evidence: (a) is this approach better than current semantic watermarks? (b) is it faster or more compute intense? (c) does it bring any further advantage to robustness over other proposed semantic watermarks?
>
> `A1:`
> (a & c)
> We clarify that the existing baseline AWTI is semantic-aware watermarking. We have expanded our baselines and evaluation as follows:
>
> (1) Additional baselines: We have now included SIR, a representative semantic watermarking method, and SynthID Text, a representative distortion-free watermarking scheme, in our comparison.
>
> (2) Expanded robustness evaluation: We further evaluate the robustness of all methods against paraphrasing attacks, including DIPPER, OPT 2.7B, and WM removal. As shown in the table below, our method demonstrates the best overall robustness.
> | Method | | No attack | | | DIPPER | | | OPT-2.7B | | | WM-removal | |
> | :--- | :---: | :---: | :---: | :---: | :---: | :---: | :---: | :---: | :---: | :---: | :---: | :---: |
> | | ROC-AUC↑ | TPR@1%FPR↑ | TPR@10%FPR↑ | ROC-AUC↑ | TPR@1%FPR↑ | TPR@10%FPR↑ | ROC-AUC↑ | TPR@1%FPR↑ | TPR@10%FPR↑ | ROC-AUC↑ | TPR@1%FPR↑ | TPR@10%FPR↑ |
> | AWTI | 1.0000 | 1.0000 | 1.0000 | 0.7391 | 0.1350 | 0.3600 | 0.9659 | 0.8400 | 0.9150 | 0.9635 | 0.7050 | 0.8450 |
> | SIR | 1.0000 | 1.0000 | 1.0000 | 0.8809 | 0.2900 | 0.8809 | 0.9623 | 0.5950 | 0.9350 | 0.9908 | 0.8550 | 0.9850 |
> | SynthID-Text | 0.9871 | 0.9400 | 0.9900 | 0.6228 | 0.0200 | 0.2700 | 0.9483 | 0.4350 | 0.8650 | 0.9772 | 0.6500 | 0.9600 |
> | **Ours** | **1.0000** | **1.0000** | **1.0000** | **0.8956** | **0.5377** | **0.7593** | **0.9931** | **0.9146** | **0.9749** | **0.9972** | **0.9598** | **0.9849** |
>
> (b) As shown in Table 3 of the main paper, our method is substantially more efficient overall. In our additional runtime evaluation of semantic watermark baselines, generating 200 watermarked texts with LLaMA takes 13.35 s/sample for our method versus 17 s/sample for SIR, while k-SemStamp requires over 24 hours under the same setting.
>
> `Q2:` What are the failure modes? What caused the detector to fail? What are potential future works to stop this from happening?
>
> `A2:` There exist three possible failure modes together with potential future work:
>
> (1) Very strong attacks: Robustness may degrade under very strong attacks. Incorporating richer contextual or sentence-level semantics could improve robustness, though at the cost of substantially higher modeling complexity (left for future work).
>
> (2) Very short texts: Performance degrades for short sequences (e.g., at $T=50$, TPR@1%FPR $≈ 0.89$; cf. Fig. 4), which is common across watermarking methods. This could be improved by designing detectors that incorporate sentence-level statistics rather than simply aggregating token-level scores.
>
> (3) SLM with mismatched tokenization: mismatched tokenization leads to inconsistencies between embedding and detection. When the source model family is unknown, a practical solution is to run detection using multiple lightweight SLMs from different candidate families. A high detection score from one SLM can both verify the watermark and indicate the most likely source model.
>
> `Q3:` Can you present some qualitative results in the appendix? If possible, can you make changes to Figure 2 to make it more intuitive?
>
> `A3:`We address your feedback as follows:
>
> (1) Qualitative examples: We will include more qualitative results in the appendix to show the watermarked texts generated by our method under different prompts.
>
> Below is a sample of watermarked text generated by our algorithm：
>
> > Check out the commercial below, featuring Conan O'Brien as the head of a company that wants to capitalize on the millions of viewers that tune in for late night shows at 3:40 AM. These kinds of advertisements are nothing new; these advertisements are backwards manipulation of associations for hidden suggestions that customers are more likely to catch than frontward association. However, by not fessing up to the truth in frontward advertisements, many companies have gained legal cause to advertise in tobacco commercial breaks, online videos of sketch comedians, and adult cartoons such as South Park, rather than the "nobody wants to hear about late night snacking" excuse.
>
> (2) Intuitive visualizations: In the final version, we will align Fig. 2 with the more intuitive teaser Fig. 1, and refine the layout and presentation, e.g., by removing heavy math notations, to make the overall pipeline easier to follow.

---

> > ### Author Rebuttal · Reviewer_oVUC · 2026-03-31
> >
> > The authors have fully addressed my concerns and I am convinced by the method and the newly presented results. The only drawback is the originality but my confidence in the necessity of the approach as well as significance would make me lean for an accept. Kudos!

---

> > > ### Author Response · Authors · 2026-04-04
> > >
> > > Dear Reviewer `oVUC`,
> > >
> > > Thank you very much for your thoughtful engagement and for taking our clarifications into consideration. We sincerely thank you for you time and feedback throughout the review process! If you find these revisions satisfactory, we would greatly appreciate your reconsideration of the current assessment.

---

### Official Review · Reviewer_EGnx · 2026-02-28

**Soundness:** 3
**Presentation:** 3
**Significance:** 3
**Originality:** 2
**Overall Recommendation:** 3
**Confidence:** 4

**Summary:**

PASA proposes a semantic-level watermarking method for LLM-generated text designed to improve robustness against paraphrasing attacks. It groups tokens into semantic clusters and embeds watermarks through a two-stage sampling process with a truncated auxiliary distribution and an overflow state, aiming to preserve the original generation distribution in expectation. Detection reconstructs the auxiliary sequence using a shared secret key and a surrogate language model. Experiments show improved robustness under paraphrasing and token replacement compared to prior methods.

**Compliance With Llm Reviewing Policy:**

Affirmed.

**Final Justification:**

The role of calibration at the detection stage remains unclear. In particular, it is not evident how the detector can be deployed efficiently in practice. For example, how many SLMs would be required to reliably cover different generation models or distributions, and what the resulting computational and calibration cost would be.

**Key Questions For Authors:**

1. Beyond introducing semantic clustering, what is the fundamental theoretical contribution compared to [1]? Can you clarify what is conceptually new rather than an extension of the existing truncation and overflow framework?

2. Can you include comparison with other semantic-based watermarking methods and results under stronger and more diverse paraphrasing models?

3. How sensitive is detection performance to mismatch between the SLM and the generation model?

**Limitations:**

See weaknesses.

**Strengths And Weaknesses:**

### Strengths

- The paper provides a natural motivation for moving from token-level watermark to a semantic-cluster framework. The experimental results show better robustness under paraphrasing compared to several prior methods.

### Weaknesses

1. Limited theoretical novelty. The theoretical framing and the distortion-free sampling (truncation + overflow) are largely extended from [1] The main difference is adding a semantic clustering layer, so the core contribution appears incremental.

2. Insufficient experimental comparison. Except for AWTI, most baselines are not semantic-based watermarking methods. Since the claimed contribution is semantic robustness, comparisons with more semantic watermarking approaches are needed, such as [2] [3]. In addition, paraphrasing evaluation mainly relies on DIPPER. The paper should include stronger and more diverse paraphrasing models (e.g., GPT4o [4]) for a more convincing evaluation.

3. Strong dependence on SLM approximation. Detection relies on a small language model (SLM) to approximate the true generation distribution. However, SLM errors can change the truncation structure, overflow mass, and the distribution of $\zeta_t$. The paper does not analyze mismatch between the SLM and the generation model or provide robustness bounds. In practice, if the generator is a domain-finetuned model, it is unclear whether a domain-specific SLM is required for detection, which raises scalability concerns.


[1] He, Haiyun, et al. "Theoretically grounded framework for llm watermarking: A distribution-adaptive approach." NeurIPS 2025

[2] Liu, Aiwei, et al. "A semantic invariant robust watermark for large language models." ICLR 2024

[3] Hou, Abe, et al. "k-SemStamp: A clustering-based semantic watermark for detection of machine-generated text." Findings of the Association for Computational Linguistics: ACL 2024.

[4] OpenAI. gpt-4o. https://platform.openai.com/docs/models/gpt-4o, 2024.

---

> ### Author Rebuttal · Authors · 2026-03-31
>
> Response to Reviewer `EGnx`: Thank you for engaging with our work in such depth! Your thoughtful questions helped clarify key aspects, and we respond point by point:
>
> `Q1:` Beyond introducing semantic clustering, what is the fundamental theoretical contribution compared to [1]? Can you clarify what is conceptually new rather than an extension of the existing truncation and overflow framework?
>
> `A1:`
> (1) Our key motivation differs fundamentally from [1]. While [1] develops a token-space theoretical framework, we instead propose a semantic embedding-space framework, explicitly bridging the gap between semantic-aware heuristic designs and token-level theory.
>
> (2) Although we adopt a similar high-level formulation as [1] up to the embedding stage (due to its generality), our framework departs critically at detection. We explicitly model semantic-invariant adversaries via a clustering function $f$ and equivalence classes $\mathcal{B}_f$, and evaluate performance under worst-case false-alarm and miss-detection probabilities. This robustness-centric criterion is absent in [1] and leads to a fundamentally different formulation.
>
> (3) Consequently, the optimal embedding and detection rules take a different form. While they may appear structurally similar to [1], our framework strictly generalizes it: when $f$ reduces to the identity mapping, we recover the classical setting without robustness.
>
> (4) Beyond theory, we propose a robust watermarking algorithm grounded in this formulation. It integrates semantic-history-based seeding and embedding-space clustering (implemented via a Qwen-based encoder), chosen for its strong semantic representation and alignment with LLM generation, enabling stable clustering under paraphrastic transformations. This design is not a direct extension of [1], but a conceptually new approach aligned with our robustness objective.
>
> `Q2:` Can you include comparison with other semantic-based watermarking methods and results under stronger and more diverse paraphrasing models?
>
> `A2:`
> We have expanded our baselines and evaluation as follows:
>
> (1) Additional baselines: We have now included SIR, a representative semantic watermarking method, and SynthID Text, a representative distortion-free watermarking scheme, in our comparison. The experiments of k-SemStamp on LLaMA is still in progress due to the extensive time required for adaptation and training. We will include the results once available.
>
> (2) Expanded robustness evaluation: We further evaluate the robustness of all methods against paraphrasing attacks, including DIPPER, OPT 2.7B, and WM removal. As shown in the table below, our method demonstrates the best overall robustness.
>
> | Method | | No attack | | | DIPPER | | | OPT-2.7B | | | WM-removal | |
> | :--- | :---: | :---: | :---: | :---: | :---: | :---: | :---: | :---: | :---: | :---: | :---: | :---: |
> | | ROC-AUC↑ | TPR@1%FPR↑ | TPR@10%FPR↑ | ROC-AUC↑ | TPR@1%FPR↑ | TPR@10%FPR↑ | ROC-AUC↑ | TPR@1%FPR↑ | TPR@10%FPR↑ | ROC-AUC↑ | TPR@1%FPR↑ | TPR@10%FPR↑ |
> | AWTI | 1.0000 | 1.0000 | 1.0000 | 0.7391 | 0.1350 | 0.3600 | 0.9659 | 0.8400 | 0.9150 | 0.9635 | 0.7050 | 0.8450 |
> | SIR | 1.0000 | 1.0000 | 1.0000 | 0.8809 | 0.2900 | 0.8809 | 0.9623 | 0.5950 | 0.9350 | 0.9908 | 0.8550 | 0.9850 |
> | SynthID-Text | 0.9871 | 0.9400 | 0.9900 | 0.6228 | 0.0200 | 0.2700 | 0.9483 | 0.4350 | 0.8650 | 0.9772 | 0.6500 | 0.9600 |
> | **Ours** | **1.0000** | **1.0000** | **1.0000** | **0.8956** | **0.5377** | **0.7593** | **0.9931** | **0.9146** | **0.9749** | **0.9972** | **0.9598** | **0.9849** |
>
> `Q3:` How sensitive is detection performance to mismatch between the SLM and the generation model?
>
> `A3:` We clarify that, in its current form, our detector is most effective when used within the same model family.
>
> (1) Detection remains highly reliable when the detecting SLM comes from the same model family as the source. Preserving the tokenizer and prediction structure is more important than model scale. As shown below, TinyLlama-1.1B [2] achieves the same perfect detection as Llama2-7B.
>
> (2) Since detection is performed at the token level, effective transfer depends on sufficient alignment in tokenization and next token prediction behavior. As a result, transfer across substantially different model families is naturally more challenging.
>
> (3) From a deployment perspective, a practical solution is to use several lightweight local detectors corresponding to different candidate families. A strong response from one detector can both verify the watermark and provide evidence about the most likely source LLM.
> | Detecting Model | ROC-AUC ↑ | TPR@1%FPR ↑ |
> | :--- | :---: | :---: |
> | Llama2-7B | 1.0000 | 1.0000 |
> | TinyLlama-1.1B | 1.0000 | 1.0000 |
>
> [1] He, Haiyun, et al. "Theoretically grounded framework for llm watermarking: A distribution-adaptive approach." NeurIPS 2025
>
> [2] Zhang P, et al. "Tinyllama: An open-source small language model"[J]. arXiv preprint arXiv:2401.02385, 2024.

---

> > ### Author Rebuttal · Reviewer_EGnx · 2026-04-03
> >
> > The SLM mismatch issue remains unclear. TinyLlama and LLaMA-2 have highly similar training distributions (mostly from C4), so the observed results may not generalize. In practice, instruction-tuned variants can differ significantly even within the same model family. If generation uses an instruction-tuned model while detection uses a base model, how does the method scale? Is per-distribution calibration required?

---

> > > ### Author Response · Authors · 2026-04-04
> > >
> > > Response to Reviewer `EGnx`:
> > >
> > > Thank you so much for your constructive and insightful comments! We address your concerns as follows:
> > >
> > > `Q1:` The SLM mismatch issue.
> > >
> > > `A1:`
> > > Thank you for the clarification. We further evaluate base/instruction-tuned mismatches within the LLaMA-2 family and per-distribution calibration is not used. The results show that detection remains effective under mismatch. Detection performance is indeed better when the generation model is the base one. When generation uses an instruction-tuned model, the detection accuracy drops slightly under either detection model. We think this is due to the fact that instruction tuning alters the next-token distribution, typically making it more structured and constrained (e.g., aligned with prompts or response styles). This reduces the effective randomness and can weaken the statistical signal exploited by the detector. In contrast, base models exhibit more stable and less constrained distributions, which are better aligned with our framework.
> > >
> > > This observation suggests several directions for improvement. In particular, incorporating lightweight per-distribution calibration and exploring the design of optimal distribution-agnostic detectors could further mitigate this gap and improve robustness under model mismatch.
> > > | Generation Model | Detector Model | ROC-AUC | TPR @ 1% FPR | TPR @ 10% FPR |
> > > |:--|:--|--:|--:|--:|
> > > | LLaMA-2-13B | LLaMA-2-7B-chat | 0.9995 | 0.9950 | 0.9950 |
> > > | LLaMA-2-13B-chat | LLaMA-2-7B | 0.9817 | 0.8750 | 0.9500 |
> > > | LLaMA-2-13B-chat | LLaMA-2-7B-chat | 0.9879 | 0.8800 | 0.9800 |
> > >
> > > We hope that our additional experiments and clarifications have addressed the reviewer’s concerns, particularly the robustness evaluations under semantic-invariant attacks. Our work aims to bridge the gap between heuristic semantic-aware methods and existing token-level theoretical frameworks. The proposed principled, robust, and distortion-free watermarking approach further demonstrates the value of theoretically grounded design.
> > >
> > > If the reviewer finds these revisions satisfactory, we would greatly appreciate a reconsideration of the current assessment.

---

### Official Review · Reviewer_EwX7 · 2026-03-11

**Soundness:** 4
**Presentation:** 4
**Significance:** 2
**Originality:** 3
**Overall Recommendation:** 4
**Confidence:** 5

**Summary:**

The authors introduce PASA, a watermark algorithm designed to be robust against semantic-invariant attacks (e.g., paraphrasing) by deriving the watermarking pseudorandom variables from a latent space representation.
Moreover, they derive their algorithm from a principled approach (they frame watermarking as an optimization problem), allowing them to obtain an a priori powerful scheme.
They evaluate their scheme on different models and datasets, against two categories of attackers (namely paraphrasing and context-aware substitution), and compare it with non-semantic watermark baselines.

**Compliance With Llm Reviewing Policy:**

Affirmed.

**Final Justification:**

Most of my concerns have been addressed I therefore recommend accepting the paper. Nonetheless, the scope of the contribution remains limited, and a few changes from the rebuttal needs to be incorporated in the next revision.

**Key Questions For Authors:**

- Q1: Do you use the prompt for detection, especially for ELI5? If not, do the authors think there could be settings in which the lack of the prompt makes detection unreliable (for instance, when some LLM-generated content is inserted in the middle of human-written text)?
- Q2: Could the authors include semantic watermark baselines, as well as SynthID?
- Q3: Could the authors include an experiment that evaluates the FPR on human-written text?
- Q4: Could the authors clarify the hyperparameters (see weaknesses)?
- Q5: Could the authors evaluate robustness against paraphrasing models from [1], which are specifically designed to break watermarks?
- Q6: Why do the authors not explore the trade-off between quality, detection, and robustness by designing a scheme that allows setting $\epsilon > 0$? This could enable a very robust scheme by trading off generation quality.
- Q7: For detection, in the case of repeated context, do you use deduplication [2]? Why or why not?

[1] https://huggingface.co/collections/DDiaa/watermark-removing-paraphrasers
[2] https://arxiv.org/abs/2308.00113

**Limitations:**

Relying on a SLM for detection can be prohibitive in some use cases. This should be stated as a limitation in the paper.

**Strengths And Weaknesses:**

**Strength**
 - The design of PASA is principled, in the sense that they derive their watermarking algorithm from an optimization problem. Despite the complexity, their notation/presentation makes the math quite easy to follow, and they rightfully defer the more complex proofs to the appendix. They also illustrate their approach extensively to ensure readers can understand it. This is a major strength of the paper.
 - They show that there is a fundamental trade-off between detection accuracy, robustness, and distortion, and characterize it through Theorem 1. The only drawback is that they do not leverage Theorem 1 to quantify how good their approach is compared to the optimal watermark.

**Weaknesses**
- The main weakness of this work is the evaluation. As it stands, I cannot decide how good this watermark algorithm is compared to prior work and whether the added complexity of their method is worth it.
	- They do not compare against prior semantic-based watermarks (like the one mentioned in related work). These would be the most relevant baselines.
	- Some hyperparameters are unclear: what is the number of tokens per generation? Do they use the prompt for the SLM at detection (especially with ELI5)? PPL should be computed with a different model than the one used for generation.
	- In Tables 1 and 2, the different rows are hard to relate because they are not at the same quality level. In particular, this does not illustrate the trade-off between quality, robustness, and detection very well. Also, the table FPR is saturated (only Figure 4 actually gives an idea of how powerful the scheme is, but without comparison to baselines).
	- For the complexity of their approach, the comparison to the baselines is slightly misleading. I assume they use the GPU for detection (whereas baselines are CPU-only). What would be more representative are the number of operations needed for detection and the memory requirements. The same applies to generation. Overall, the main weakness with regard to complexity is that detection appears to require an LLM of similar size to the one used for generation.
	- SynthID should be used as a baseline, as it is a very popular distortion-free scheme.
- While their approach offers better robustness to paraphrasing, the actual TPR remains low for practical use. What would be the number of tokens needed to achieve 99 TPR@1% FPR under paraphrasing with their approach? Also, robustness to context-aware substitution seems on par with some other baselines.
- The authors should include an experiment that shows the empirical FPR with respect to alpha when running detection on human-generated text. In some prior works, because of a lack of "true randomness" in their pseudo-randomness generation, the empirical FPR given a fixed watermarking key could be off by a significant margin. I wonder if this is the case here, given the complex pipeline used to generate the $\zeta$.
- Despite a long introduction, it lacks motivation for why a watermark robust to semantic attacks is needed, and what costs are acceptable to achieve strong robustness (e.g., with the proposed scheme, detection costs increase).

---

> ### Author Rebuttal · Authors · 2026-03-31
>
> Dear Reviewer EwX7,
>
> Thank you so much for your constructive and insightful comments! We address your concerns point by point as follows.
>
> `Q1:` Prompt for detection, especially for ELI5?
> `A1:` We clarify our detection settings as follows:
>
> - **Detection without prompts:** In all experiments (both **C4** and **ELI5**), detection is performed without prompt. We have added explicit results to make this setting clear.
>
> - **Mixed-text setting:** We have not yet systematically studied the more challenging case where human written text is inserted into the middle of a watermarked passage. As a preliminary step, we evaluate a setting where the human-written prompt is prepended to the watermarked text. The table below shows that the watermark remains reliably detectable in this case.
>
> More general mixed-text scenarios are left for future work.
>
> Detection performance **with** and **without** prompts below:
>
> | Setting             | C4 ROC-AUC | C4 TPR @ 1% FPR | C4 TPR @ 10% FPR | ELI5 ROC-AUC | ELI5 TPR @ 1% FPR | ELI5 TPR @ 10% FPR |
> |:--------------------|-----------:|----------------:|-----------------:|-------------:|------------------:|-------------------:|
> | **Without prompt**  |     1.0000 |          1.0000 |           1.0000 |       1.0000 |            1.0000 |             1.0000 |
> | **Mixed with prompt** |   0.9997 |          0.9899 |           1.0000 |       1.0000 |            1.0000 |             1.0000 |
>
> `Q2 & Q5:` More baselines and paraphrasing models?
>
> `A2 & A5:`
> We have expanded our baselines and evaluation as follows:
>
> (1) We have now included SIR [1] (semantic-aware) and SynthID (distortion-free) in our comparison.
>
> (2) Additional robustness evaluation: We further evaluate the robustness of all methods against more paraphrasing attacks. As shown in the table below, DIPPER attack is indeed stronger than other attacks and our method demonstrates the best overall robustness. ***Due to limited space, we kindly ask the reviewer to refer to the table in response to Reviewer `oVUC`.***
>
> `Q3 & Q9:` FPR on human-written text? Empirical FPR vs $\alpha$?
>
> `A3 & A9:` In all our detection experiments, human-written text is treated as "unwatermarked text", and the reported FPR is precisely computed on human-written data. We will clarify this later in the paper to avoid confusion.
>
> The table below shows the detection performance on 200 samples under different $\alpha$. In theory, $\alpha$ controls the worst-case false alarm error at each position and may be conservative compared to the empirical FPR, e.g.FPR@95%TPR=0 for all $\alpha$.
> | Metric |$\alpha=$ 0.1 | 0.2 | 0.3 | 0.4 | 0.5 | 0.6 | 0.7 | 0.8 | 0.9 | 1.0 |
> |:--|--:|--:|--:|--:|--:|--:|--:|--:|--:|--:|
> | ROC-AUC | 0.9931 | 0.9937 | 0.9970 | 0.9999 | 1.0000 | 1.0000 | 1.0000 | 0.9998 | 0.9988 | 0.9967 |
> | TPR @ 1% FPR | 0.9146 | 0.9700 | 0.9950 | 0.9950 | 1.0000 | 1.0000 | 1.0000 | 1.0000 | 0.9900 | 0.9950 |
> | TPR @ 10% FPR | 0.9749 | 0.9850 | 0.9950 | 1.0000 | 1.0000 | 1.0000 | 1.0000 | 1.0000 | 1.0000 | 1.0000 |
>
> `Q4:` Clarify the hyperparameters (see weaknesses)? PPL evaluation?
>
> `A4:` Unless otherwise specified, each sample is generated with 200 to 300 tokens; Additional results under varying lengths are provided in Appendix A.
>
> We add new PPL evaluations under OPT-13B:
> | Metric | Human | Unwatermark | EXP | AWTI | SIR | OURS |
> | :--- | :--- | :--- | :--- | :--- | :--- | :--- |
> | **PPL** | 10.21 | 11.85 | 22.64 | 11.71 | 14.46 | 14.04 |
>
> `Q6:` Why not allow \epsilon > 0 in the scheme?
>
> `A6:` Our theory already characterizes the full trade-off via the distortion parameter $\epsilon$. In the algorithm, we focus on the distortion-free regime ($\epsilon = 0$) to preserve generation quality.
>
> Our results show that strong robustness and reliable detection can be achieved **without introducing distortion**. While allowing $\epsilon > 0$ is compatible and can further improve robustness, we deliberately focus on $\epsilon = 0$ to highlight this more stringent result, distinguishing our approach from prior methods that rely on explicit distortion. We leave further improvement for future work.
>
> `Q7:` Deduplication [2] in detection?
>
> `A7:` We confirm that we do not use deduplication for our method or any of the reproduced baselines. To ensure a fair and realistic evaluation, all detection methods are applied directly to the generated text without any additional processing.
>
> `Q8:` Tables 1 and 2 are hard to relate.
>
> `A8:` Tables 1 and 2 correspond to different experimental settings, rather than matched comparisons at the same quality level. Table 1 reports results under T5-based attacks for both Llama2-13B and Mistral-8×7B, while Table 2 focuses on DIPPER attacks for Llama2-13B only.
>
> Thus, only the Llama2-13B results are directly comparable across the two tables, whereas the Mistral-8×7B rows are not.
>
> [1] Liu, Aiwei, et al. "A semantic invariant robust watermark for large language models." ICLR 2024

---

> > ### Author Rebuttal · Reviewer_EwX7 · 2026-04-01
> >
> > I thank the authors for their rebuttal! Some of my points were addressed, but not all of them. I list below the concerns that are still not fully addressed for clarity.
> >
> > **Q1**
> > - The mixed-text setting is actually very relevant and I believe should not be left for future work. I doubt there would be any future works doing this kind of experiments.
> >
> > **Q3 \& Q9**
> > - I do not understand what you provided. Can you simply run the detection on any human text and report the percentage of human text classified as watermarked for different values of $\alpha$? Or explain in more depth the table you provided.
> >
> > **Q4**
> > - Why is the PPL (significantly?) higher than that of human text for your scheme, given that it is distortion-free?
> >
> > **Q8**
> > - Sorry for the misunderstanding. I meant that the rows within the table are hard to relate to one another (see the original comment in Weaknesses).
> >
> > **Others**
> > I think these points are still not fully addressed:
> > - For Table 3, "I assume they use the GPU for detection (whereas the baselines are CPU-only). What would be more representative are the number of operations needed for detection and the memory requirements. The same applies to generation. Overall, the main weakness with regard to complexity is that detection appears to require an LLM of similar size to the one used for generation.''
> > - "What would be the number of tokens needed to achieve 99 TPR@1\% FPR under paraphrasing with their approach?''
> > - Lack of motivation in the introduction.

---

> > > ### Author Response · Authors · 2026-04-04
> > >
> > > Response to Reviewer `EwX7`:
> > >
> > > Thank you for your follow-up. We address your concerns as follows:
> > >
> > > `Q1:` Mixed Setting.
> > >
> > > `A1:`
> > > We have now included this experiment in the revision. Even with human-written text inserted into watermarked text, our method maintains strong detection performance.
> > > Detection performance below:
> > > |Human insert ratio (%)|ROC-AUC|TPR @ 1% FPR|TPR @ 10% FPR|
> > > |-|-|-|-|
> > > |15|0.9996|0.9798|1.0000|
> > > |30|0.9981|0.9296|1.0000|
> > > |50|0.9915|0.8492|0.9748|
> > >
> > > `Q3 & Q9:` Clarification of the table and FPR on human text vs. $\alpha$.
> > >
> > > `Q3 & Q9:`
> > > As $\alpha$ in theory is the worst-case FPR upper bound at each position, the empirical FPR of a sequence would be far below $\alpha$, which implies that $\alpha$ effectively controls empirical FPR. The table below reports the FPR on $500$ human-written texts (token length$\approx$300) at TPR=95%, under different $\alpha$. The reported FPRs are indeed far below $\alpha$ and very close to $0$.
> > > |Alpha|0.1|0.2|0.3|0.4|0.5|0.6|0.7|0.8|0.9|
> > > |:-|:-:|:-:|:-:|:-:|:-:|:-:|:-:|:-:|:-:|
> > > |FPR@TPR=95%|0.8%|0.0%|0.0%|0.0%|0.0%|0.0%|0.2%|0.4%|0.4%|
> > >
> > > `Q4:` Why is the PPL (significantly?)
> > >
> > > `Q4:`
> > > We have added new PPL evaluations using multiple external language models, including Qwen2.5-7B, GPT-NeoX-20B, and OPT. The results are shown below:
> > > |Metric|Human|Unwatermark|EXP|AWTI|SIR|OURS|
> > > |-|-|-|-|-|-|-|
> > > |Qwen|10.88|13.46|26.26|20.60|14.75|12.74|
> > > |GPT|10.41|12.41|23.40|19.77|15.15|11.44|
> > > |OPT|10.21|11.85|22.64|11.71|14.46|14.04|
> > >
> > > These results show that while absolute PPL values vary across evaluator models (as expected), our method consistently remains much closer to human text than other baselines.
> > >
> > > `Q8:` The rows within the table are difficult to compare.
> > >
> > > `Q8:`
> > >  Within each table, different watermarking methods indeed generate watermarked text with different qualities in practice. However, we know that KGW+23 and AWTI are **distorted** designs, while EXP-edit and DAWA are **distortion-free** designs. Therefore, we aim to compare our distortion-free design with these two sets of baselines and show that we achieve both high detectability and robustness at the same time. We will inclue the new baselines (SynthID and) in our rebuttal to the final paper and illustrate the three-dimensional trade-off in a clearer way.
> > >
> > > As we do not discuss explicity about the effect of token length, Fig. 4 in appendix is not our main result. We will add more baselines in Fig. 4 in the final paper and we expect to see similar results as those in Tables 1&2.
> > >
> > > `Q10:` Detection operations and memory requirements.
> > >
> > > `Q10:`
> > > We now present memory requirements and computational cost for both generation and detection. The results show that our method do not require much higher complexity than other semantic-aware methods.
> > >
> > > For generation, all methods use the same generation backbone, which requires 25,376 MB under our setup. Thus, the generation-side memory requirement is essentially the same across methods.
> > >
> > > For detection, the costs are summarized below:
> > > ||DAWA|EXP|AWTI|SIR|OURS|
> > > |-|-|-|-|-|-|
> > > |Memory(MB)|2892|CPU|6344|1994|2892|
> > > |GFLOPs|1.06×10³|5.12×10⁻⁵|2.85×10⁵|1.32×10⁴|6.21×10²|
> > >
> > > `Q11:` Required tokens for 99% TPR@1% FPR (paraphrased).
> > >
> > > `Q11:`
> > > Achieving 99%TPR@1% FPR is essentially hard even in the clean text setting. Due to the limited rebuttal time, we have included additional experiments to evaluate detection performance under paraphrasing (Dipper with Lex=20 and Ord=0) across several token lengths, as shown below:
> > > |Max Token Length|ROC-AUC|TPR @ 1% FPR|TPR @ 10% FPR|
> > > |-|-|-|-|
> > > |450|0.9895|0.9196|0.9849|
> > > |600|0.9929|0.9296|0.9749|
> > > |800|0.9898|0.9347|0.9899|
> > > |2000|0.9955|0.9558|0.9912|
> > >
> > > The results demonstrate that performance improves significantly with longer token sequences, substantially outperforming the results reported in Table 2 for the 200–300 token range. In particular, we achieve up to 99% ROC-AUC and 99% TPR at 10% FPR.
> > >
> > > In the final version, we will include corresponding baseline results in this table for a more comprehensive comparison.
> > >
> > > `Q12:`  Lack of motivation in the introduction.
> > >
> > > `A12:`
> > > On Page 2 (right column Lines 58–70), we outline our motivation. In the revised version, we will more clearly state the target problem, limitations of existing methods, and the necessity of our approach, with a stronger connection between practical challenges and our technical design.
> > >
> > > Our motivation stems from a fundamental gap between practice and theory in robust LLM watermarking. Practically, semantic-aware methods suggest that operating in embedding space improves robustness to semantic-invariant attacks. However, theoretically, existing frameworks are largely limited to token-level watermarking and fail to account for meaning-preserving transformations, leaving trade-offs among robustness, distortion, and detection accuracy poorly understood.
> > >
> > > This gap motivates our work: a principled, semantic-level watermarking framework with explicit theoretical guarantees.

---

### Official Review · Reviewer_fN1P · 2026-03-17

**Soundness:** 3
**Presentation:** 3
**Significance:** 3
**Originality:** 3
**Overall Recommendation:** 4
**Confidence:** 3

**Summary:**

The paper proposes PASA, a novel watermarking framework designed to be robust against semantic-invariant attacks such as token replacement or paraphrasing. Different from existing token-level methods, PASA operates in the semantic space. It pre-clusters tokens into semantic clusters using an embedding model. During watermark embedding, it uses a secret key combined with the semantic history to decide which semantic cluster the next token should be sampled from, rather than operating on individual tokens. During detection, it uses the same secret key and the clusters, but uses a smaller proxy model to approximate the prediction probability distribution. Experimental results show that PASA outperforms previous strong baselines.

**Compliance With Llm Reviewing Policy:**

Affirmed.

**Key Questions For Authors:**

can the authors address the points raised in the weakness part?

**Limitations:**

no explicit discussions of limitations.

**Strengths And Weaknesses:**

Strengths:
1. The paper provides a theoretical proof about trade-offs between detection, robustness and distortion. It then derives a jointly optimal embedding and distortion-free scheme, and instantiates this proof in the proposed method.
2. The experimental results show that PASA achieves strong results compared to baseline methods, especially under the paraphrasing setting, where token-level baselines degrade significantly.
3. The method preserves the original prediction distribution, validated by the perplexity scores that closely match the unwatermarked baseline.

Weaknesses:
1. The method relies on the prediction distribution of the small models used in the detection phase aligning with the watermarked source model. This paper uses the smaller variants of the same model family. How robust is the detection when using different SLMs, especially when the source model is unknown?
2. The details of the token embedding encoding are slightly unclear. For instance, what instruction was used? Also, encoding static single token embeddings may not be optimal as it doesn’t encode contextualized semantics of words.
3. The method seems to be highly sensitive to the number of clusters K. This suggests that the method might require heavy tuning for different models or languages, and might be sensitive to the embedding models used.

---

> ### Author Rebuttal · Authors · 2026-03-31
>
> Dear Reviewer fN1P,
>
> Thank you sincerely for your valuable comments and engagement throughout the review process. We hope our responses have addressed your concerns.
>
> `Q1:` How robust is the detection when using different SLMs, especially when the source model is unknown?
>
> `A1:`
> We clarify that, in its current form, our detector is most effective when used within the same model family sharing identical tokenization.
>
> (1) **Same tokenization:** Detection remains highly reliable when the SLM shares the same tokenization as the source model. As shown below, TinyLlama-1.1B [1] achieves the same perfect detection as Llama2-7B. This is because mismatched tokenization leads to inconsistencies between embedding and detection. In the future, we will study how to mitigate the effects of different tokenizations.
>
> (2) **Unknown source model:** When the source model is unknown, a practical solution is to run detection using multiple lightweight SLMs from different candidate families. A high detection score from one SLM can both verify the watermark and indicate the most likely source model.
>
> We leave improving robustness across different tokenizations as future work.
>
> | Detecting Model | ROC-AUC | TPR @ 1% FPR |
> |:--|--:|--:|
> | Llama2-7B | 1.0000 | 1.0000 |
> | TinyLlama-1.1B | 1.0000 | 1.0000 |
>
> `Q2:` For instance, what instruction was used?
>
> `A2:`
> We clarify our approach as follows:
>
> (1) Instruction used: We use gte-Qwen2-7B-instruct with a fixed instruction along the lines of "capture the semantic function of this vocabulary item" to encode each token individually for clustering. The model encodes token semantics in a context-aware manner.
>
> (2) Why token-level semantics: Although our theoretical foundations give an optimal embedding-detection pair for sequences, our algorithm operates at the token level due to the computational complexity of working in the probability simplex at the sentence level. We hope to achieve a balance between theoretical optimality and computational efficiency. Our experiments show that our algorithm is both effective and efficient.
>
> We agree that incorporating richer contextual or sentence-level semantics could further improve the method, but this would introduce substantially higher modeling complexity and is left for future work.
>
> `Q3:` The method seems to be highly sensitive to the number of clusters K.
>
> `A3:` We clarify that semantic clustering is implemented via a fixed, independent model and depends only on the tokenization of the watermarked model. For models sharing the same tokenization, \(K\) does not need to be re-tuned; for different tokenizations, clustering (and \(K\)) is reselected, but this process is lightweight.
>
> Importantly, the sensitivity to \(K\) is limited in practice:
> - **Narrow effective range:** Empirically, \(K\) performs well within a small range (typically 3–6), so extensive tuning is unnecessary.
> - **Role of \(K\):** Our theory shows that \(K\) primarily controls robustness—-smaller \(K\) yields stronger robustness under semantic-invariant attacks. It does not affect clean-text detection performance (cf. Fig. 3).
> - **Quality impact:** Not intrinsic to the theory, but due to semantic-history-based seeding. Empirically, the following table shows that the effect on PPL is relatively small.
>
> Overall, tuning \(K\) within a small range is sufficient and does not pose a practical burden.
>
> | K | 2 | 3 | 4 | 6 | 12 | 30 |
> | :--- | :---: | :---: | :---: | :---: | :---: | :---: |
> | wm_PPL ↓ | 11.2186 | 11.3903 | 10.5023 | 12.7979 | 11.0367 | 10.1628 |
>
> [1] Zhang P, et al. "Tinyllama: An open-source small language model"[J]. arXiv preprint arXiv:2401.02385, 2024.

---

> > ### Author Rebuttal · Reviewer_fN1P · 2026-04-06
> >
> > Thanks for clarification about embedding instructions and clustering size. The concern about the effectiveness of the method when different source and detection models are used (although the authors show that it works for models with same tokenization), especially when the source model is unknown, remains not fully addressed beyond explanation. I keep my current rating of weak accept.

---

> > > ### Author Response · Authors · 2026-04-08
> > >
> > > Thank you for the insightful feedback.
> > >
> > > The main focus of this work is to propose a theoretically guided robust semantic-level watermarking method, with the primary goal of improving robustness against semantic-invariant attacks. While the use of SLMs does introduce some limitations in deployment, our method still shows cross-SLM detection transferability to some extent, as shown in the table above. We do not claim to address the fully unknown source-model case in this paper, and we leave this to future work.
> > >
> > > | Detecting Model | ROC-AUC | TPR @ 1% FPR |
> > > |:--|--:|--:|
> > > | Llama2-7B | 1.0000 | 1.0000 |
> > > | TinyLlama-1.1B | 1.0000 | 1.0000 |
> > >
> > > We also evaluate base/instruction-tuned mismatches within the LLaMA-2 family, where the chat model is an instruction-tuned variant of the base model. The results show that detection remains effective, with only limited performance drop.
> > >
> > > | Generation Model | Detector Model | ROC-AUC | TPR @ 1% FPR | TPR @ 10% FPR |
> > > |:--|:--|--:|--:|--:|
> > > | LLaMA-2-13B | LLaMA-2-7B-chat | 0.9995 | 0.9950 | 0.9950 |
> > > | LLaMA-2-13B-chat | LLaMA-2-7B | 0.9817 | 0.8750 | 0.9500 |
> > > | LLaMA-2-13B-chat | LLaMA-2-7B-chat | 0.9879 | 0.8800 | 0.9800 |
> > >
> > > Beyond the strong detection performance reported in the main text, our current method still offers strong practical value, achieving significantly lower memory usage and favorable computational cost when compared with representative baselines.
> > >
> > > ||DAWA|EXP|AWTI|SIR|OURS|
> > > |-|-|-|-|-|-|
> > > |Memory(MB)|2892|CPU|6344|1994|2892|
> > > |GFLOPs|1.06×10³|5.12×10⁻⁵|2.85×10⁵|1.32×10⁴|6.21×10²|

---

### Decision · Program_Chairs · 2026-04-30

**Decision:**

Accept (regular)

**Comment:**

The paper proposes a semantic LLM watermarking method with a theoretical analysis of the detection/robustness/distortion trade-offs. Three reviewers lean toward acceptance, and one recommends weak rejection. The rebuttal period added useful material, including new baselines (SIR and SynthID), calibration, and FPR checks on human text. Two issues remain after the rebuttal: (1) the theoretical contribution over He et al. (NeurIPS 2025) is thin, and (2) the detector depends on a surrogate SLM that must be close to the generator, and under mismatch, the rebuttal shows a measurable drop in detectability. Robustness is also not especially high, since the authors only reach 0.54 TPR@1%FPR under DIPPER paraphrasing.

  That said, the initial weaknesses raised by the reviewers were evaluation breadth, and the rebuttal fixed most of them. SIR and SynthID were added, and PASA outperforms them on DIPPER by a large margin, with similar gaps on OPT-2.7B and watermark-removal attacks. PPL across three external evaluators stays close to human text, which fN1P specifically called out as validating the distortion-free claim. EwX7 and oVUC
  moved to accept after the rebuttal. EGnx held at weak reject over the same-family SLM requirement, which is a fair concern, but even EGnx scored the paper's soundness, presentation, and significance as good. Overall, I recommend weak accept.